# Novel coupled permafrost-forest model (LAVESI-CryoGrid v1.0) revealing the interplay between permafrost, vegetation, and climate across eastern Siberia

Stefan Kruse[1*], Simone M. Stuenzi[1,2], Julia Boike[1,2], Moritz Langer[1,2], Josias Gloy[1], Ulrike Herzschuh[1,3,4]

1 Alfred Wegener Institute Helmholtz Centre for Polar and Marine Research, 14473 Potsdam, Germany
2 Department of Geography, Humboldt Universität zu Berlin, 12489 Berlin, Germany
3 Institute of Environmental Sciences and Geography, University of Potsdam, 14476 Potsdam, Germany, and
4 Institute of Biochemistry and Biology, University of Potsdam, 14476 Potsdam, Germany

Correspondence to Stefan Kruse: stefan.kruse@awi.de

**Abstract.** Boreal forests of Siberia play a relevant role in the global carbon cycle. However, global warming threatens the existence of summergreen larch-dominated ecosystems likely enabling a transition to evergreen tree taxa with deeper active

layers. Complex permafrost-vegetation interactions make it uncertain whether these ecosystems could develop into a carbon source rather than continuing atmospheric carbon sequestration under global warming. Consequently, shedding light on the role of current and future active-layer dynamics and the feedbacks with the apparent tree species is crucial to predict boreal forest transition dynamics, and thus for aboveground forest biomass and carbon stock developments. Hence, we established a coupled model version amalgamating a one-dimensional permafrost-multilayer forest land-surface model (CryoGrid), with

LAVESI, an individual-based and spatially explicit forest model for larch species (*Larix* Mill.), extended for this study by including other relevant Siberian forest species and explicit terrain.

Following parametrization, we ran simulations with the coupled version to the near future to 2030 with a mild climate-warming scenario. We focus on three regions, covering a gradient of summergreen forests in the east at Spasskaya Pad to mixed summergreen-evergreen forests close to Nyurba, and the warmest area at Lake Khamra in the south-east of Yakutia, Russia.

Coupled simulations were run with the newly implemented boreal forest species and compared to runs allowing only one species at a time, as well as to simulations using just LAVESI. Results reveal that the coupled version corrects for overestimation of active-layer thickness (ALT) and soil moisture and large differences in established forests are simulated. We conclude that the coupled version can simulate the complex environment of Eastern Siberia reproducing vegetation patterns making it an excellent tool to disentangle processes driving boreal forest dynamics.

## 1 Introduction

Boreal forests cover vast areas of the northern hemisphere with strong gradients in climatic conditions and environments. They established in the northern hemisphere after the last glacial maximum, leaving only a thin stretch in the north bordering the

Arctic Ocean of pristine tundra areas (Mamet et al., 2019, Bonan, 2008, MacDonald et al., 2010). Anthropogenic climate warming is leading to the relaxation of warmth-deficit limits at the northern margins and hence invasion of the tundra at a yet unclear rate (Berner, 2013, Reese et al., 2020). At the same time, large parts of boreal forests, especially in Eastern Siberia, are exposed to increasing disturbances (such as fires and drought) potentially driving a forest transition from deciduous species to evergreen taxa (Bonan, 2008, Herzschuh 2020). Accordingly, wildfires lead to increased greenhouse gas emissions through burning biomass and by deepening of the seasonally thawed layer for decades. The forest transition furthermore reduces the albedo leading to a net positive global warming feedback, which will likely not be offset by increased carbon sequestration of a denser understory vegetation (Bonan, 2008). However, the involved forest dynamics and interactions with the atmosphere and soil need to be considered in sufficient detail to forecast more realistic projections and to better understand the consequences for the boreal permafrost ecosystems of Siberia (Kirpotin et al., 2021).

Forest modelling is typically done globally including the carbon cycle/permafrost etc. but individual trees and all life-history stages need to be considered for a precise simulation. Modern global models such as LPJ-GUESS (Zhang et al., 2013) include individual models. The global models are used to show that forests will change and advance north. However, migration lags are typically not represented and only climate envelopes serve for the distribution of plant functional types (PFTs). Dispersal processes and complexities have recently been recognized (Snell 2014, Snell & Cowling 2015, Lehsten et al., 2019) but are not yet used as standard for simulations. Furthermore, most modelling schemes still start with established trees, which makes them more general and computationally effective for a global application but at the cost of losing important detail for ecosystem responses (see discussion in Kruse et al., 2016). Also, the use of representative grid cells on a large grid without considering landscape will cause deviations to an extent that is unclear as to whether the impact on results is large and significant or not. Individual based models (IBMs) could help here as they have sufficient detail of represented species/ecosystems but applications are therefore only possible on landscapes not continents (Grimm & Railsback, 2005, DeAngelis & Mooij, 2005). Nevertheless, IBMs are the best tools to understand a system and develop general responses that can then inform or guide global model development. Further, neither a radiative transfer scheme through a multilayer canopy nor detailed representation of permafrost are included in typical simulation approaches.

Here, we aim at creating a model system that can accurately assess detailed thermal and hydrological fluxes between permafrost and forest cover as recently developed by Stuenzi et al. (2021a, 2021b). The new model will include a dynamic vegetation model, which has a full life-cycle to allow intraspecies and interspecies interactions at all stages (seed–seedling–mature tree) leading to non-linear behaviour of population dynamics as well as resolving a 3-dimensional landscape that is available for Siberian treeline areas developed by Kruse et al. (2016, 2019a, 2019b).

## 2 Methods

We further developed two models and start each description by using the Overview part of the ODD protocol for describing individual based models (Grimm et al, 2010). We describe the host model LAVESI in section 2.1, a spatially-explicit,

individual-based model handling the full life-cycle of tree species and interactions among individuals and its environment. The second model CryoGrid, which is informed by the host model and delivers improved state variables back to LAVESI, is described in section 2.2, a one-dimensional, numerical land surface model that simulates the thermo-hydrological regime of permafrost ground by numerically solving the heat-conduction equation.

## 2.1 The 2D vegetation model LAVESI

### 2.1.1 Model description

The *Larix* vegetation simulator (LAVESI) is an individual-based spatially explicit model that simulates larch stand dynamics (Kruse et al, 2016; Kruse et al, 2018). Monthly temperatures of the coldest (January) and warmest (July) months and precipitation series can force this model. In addition, 6-hourly data on wind speed and direction are needed to simulate seed distribution and tree reproduction, growth, and death (Kruse et al., 2019b; Kruse et al., 2018; Kruse et al., 2016). Recently the model has been extended by including topography and landscape sensing of the individuals (section 2.1.2) and further boreal forest tree species were introduced aside from the larch species the model was initially developed for (section 2.1.3; see full changes in Appendix A).

#### 2.1.1.1. Purpose

The novel coupled model LAVESI-CryoGrid v1.0 was set up to understand tree stand structure, migration and population dynamics of boreal forests growing between the leading edge at the Siberian treeline ecotone and the southern limit in response to a changing climate and its feedbacks with permafrost soils.

#### 2.1.1.2. Entities, state variables, and scales

The model consists of two hierarchical levels characterized by a set of variables (Table 1): (1) simulation areas characterized by the specific biotic and abiotic environment, and (2) individual trees and seeds.

The individual simulation areas are variable and have a size of typically 510x510 m (for parameterization and simulation experiments) on which seeds and trees are exactly positioned by *x,y* coordinates. Using the basal diameter of individual trees, the plot is overlaid with a tree density grid with a resolution of 0.2x0.2 m.

Simulation runs proceed in yearly time steps. We performed simulations for years 1–2100, prolonged by RCP prediction scenarios. Additionally, to reach stabilization of population dynamics and the forcing climate series, simulations were preceded by a stabilization period with a length of 1,000 years (for parameterization and sensitivity analysis). All simulations start from bare ground introducing 5000 ha$^{-1}$ yr$^{-1}$ seeds in the first 50 years and, to allow for repopulation of simulation areas after extinction, 100 ha$^{-1}$ yr$^{-1}$ seeds are added every year to the simulation areas.

### 2.1.1.3. Process overview and scheduling

The simulation proceeds in yearly time steps from the beginning to the end of the input climate time-series, which includes a stabilization period to ensure that emerging populations reach equilibrium with the environment. In each initialization phase of each simulation run, the weather data are processed and used to estimate maximum diameter growth (at basal and breast height) for each simulation year based on 10-years mean climate auxiliary variables (see details in '2.2.2 Description of sub-models' in Kruse et al., 2016). Within the growth processes of the model, these variables are used to individually estimate the current diameter growth of trees constrained by their actual biotic (competition) and abiotic (landscape features: elevation, TWI, slope, soil moisture, active layer depth) environment (Design concept: Sensing). Stochasticity in the model was introduced by using random numbers generated with a pseudo random number generator (mt19937_64, from the random library) to allow for different results between two or more consecutive runs of the model; Design Concept: Stochasticity).

Within one simulation year, the following processes become consecutively invoked (see Fig. 2 in Kruse et al. (2016), and for detailed explanations for each process can be found in a corresponding section in '2.2.2 Description of sub-models'): **Update of environment:** Interactions between neighbouring trees are local and indirect. Basal diameters of each individual tree are used to evaluate the competition strength. We use a yearly updated density map to pass information about competition for resources between trees. (Design Concept: Interaction). Further, a litter layer and the state variables of each grid cell are updated as well. **Growth:** The individual growth of basal diameter and, if a tree reached a height of 1.3 m, of breast height diameter, is calculated from the maximum possible growth in the current year affected by the tree's density index and its abiotic environment. From the resulting diameters, the tree height is estimated differently for the two height classes, smaller and greater than 1.3 m. (Design Concept: Collectives). **Seed dispersal:** Seeds in 'cones' are dispersed from the parent trees, at a set rate. The dispersal directions and distances are randomly determined from a ballistic flight influenced by wind speed and direction with decreasing probabilities for long distances and only to places lower than the release height. If dispersed seeds leave the extent of the simulated plot they are removed from the system, but optionally they could be introduced from the other side or only on the east-west margins, depending on the user's choice. **Seed production:** Trees produce seeds after the year at which they reached their stochastically estimated maturation height. The total amount depends on weather, competition, and tree size. Optionally, the pollen donor for the pollination of ovules of seeds produced can be selected by a wind-determined and distance-dependent probability distribution function using a von Mises distribution. **Establishment:** The seeds that lie on the ground germinate at a rate depending on current weather conditions and is constrained by the actual litter layer height. **Mortality:** Individual trees or seeds die, i.e. they become removed from the plot, at a specified mortality rate. For trees this is deduced from long-term mean weather values, a drought index, surrounding tree density, tree age and size, plus a background mortality rate. Seeds on the other hand have the same constant mortality rate whether on trees and or the ground. (Design Concept: Emergence). **Ageing:** Finally, the age of seeds and trees increases once a year and seeds are removed from the system when they reach a defined species age limit.

### 2.1.2 Addition of landscape sensing

Data from the digital elevation model (DEM) TanDEM-X 90 m was downloaded from the web service provided by the German Aerospace Center (DLR https://download.geoservice.dlr.de/TDM90/; Krieger et al., 2007). Subsequently, the tiles were reprojected to the corresponding UTM zone of the focus areas (Khamra N49, Nyurba N50, Spasskaya Pad N52). All tiles were merged for each subzone and resampled by linear interpolation from 90 m to 30 m resolution using functions from the "raster" package in R (Hijmans, 2020). The results were imported in SAGA GIS version 2.3.2 (Conrad et al., 2015) and subjected to a basic terrain analysis tool using the standard parameters. The resulting rasters were water masked using the cloud-based geospatial data analysis platform Google Earth Engine (GEE, Gorelick et al., 2017) to assess Sentinel-2 imagery between 1st May 2018 and 15th October 2018 with a cloud cover of less than 20% and thresholds manually set for spectral band B12 (2190 nm) until all water was masked out by comparing them to an RGB composite image. The DEM along with slope angle and terrain water index (TWI, moisture content) were cropped to 510x510 m (260,100 m²) areas for this study and exported as plain text files for import into LAVESI.

LAVESI reads this data provided in 30 m resolution and interpolates linearly from the closest four grid cells for each 20 cm grid tile of the environment grid. Based on empirical relationships of forest presence for combinations of slope angle and TWI established in a study by Shevtsova et al. (in review, 2021), an environment growth impact factor (*Envirgrowth*, 0–1) is calculated for each tile and tree diameter growth at this position is reduced accordingly (Eq. 1, Appendix B).

$$Envirgrowth_i = \frac{-0.045999 * TWI_i + 0.994066}{2} + \frac{0.85654 * e^{-0.5 * (Slope_i - 8.78692)^2 / 6.90743^2}}{2}, \qquad (1)$$

where $TWI_i$ is the interpolated terrain water index of the 20x20 cm² environmental grid cell $i$, and $Slope_i$ is the slope angle of the same grid cell $i$.

Seed dispersal has been improved. Seeds can now only be dispersed to places which are at the same or lower elevation than the release height in the terrain.

### 2.1.3 Addition of species and estimating leaf area index (LAI)

Further species were added to the existing model presented in Kruse et al. (2016). To add a fast forward implementation of species in LAVESI, we modified the code so that the program can be started with either one or all species in a mix simultaneously. The species are numbered (integer values), which are used internally to assess species-related variables (Table 2) when called for in the functions as necessary. Therefore, the code is independent from the species and allows adding species or functional types simply by adding a new line in the new *specieslist.csv* in the main folder of LAVESI.

For this study, we analysed field data from the Chukotka and central Yakutia 2018 expedition in the same way as we did for Chukotka (Biskaborn et al., 2019, Kruse et al., 2019a). In the area of central Yakutia, species belonging to the Pinaceae family form the forests. From these, two deciduous boreal forest tree species were sampled, *Larix cajanderi* Mayr. (LACA) and *Larix*

*gmelinii* (Rupr.) Rupr.*,* (LAGM), and three evergreen species, *Picea obovata* Ledeb. (PIOB), *Pinus sibirica* Du Tour (PISI), and *Pinus sylvestris L.* (PISY) (Kruse et al., 2019a). While the two larch species are best adjusted to the harsh environment of Northeast Siberia, and are able to grow on shallow active layers above permafrost, they differ mainly in their frost hardiness and the species LACA can even endure colder temperatures in winter (Table 2). PIOB is a competitor for *L. gmelinii* growing at similar environmental conditions, however preferring deeper thawed active layers of minimum of 200 cm. On well-drained sites, PISY grows well and outcompetes the other species. In milder environments, LASI and PISI grow on similar sites as LAGM and PIOB.

Tree-ring width data were established from tree discs and cores collected from sites close to Lake Khamra and from the region Nyurba. The discs and tree cores were prepared by standard dendroecological processing steps: (1) sanding with progressively finer paper until tree rings are clearly visible, (2) making high-resolution images for a track with a binocular and attached camera, (3) detecting rings with CooRecorder (Cybis Elektronik & Data AB) and cross-dating, and (4) exporting individual tree-ring chronologies (more details in Kruse et al., 2020). Tree-ring width data per species were then imported to R using the dplR package (Bunn et al., 2020) and regression models were set up by fitting nonlinear functions using generalized least squares with the gnls-function from the nlme package (Pinheiro et al., 2019). For each species, we extracted the median of the loess-smoothed (span=1.5) yearly growth increase of individual trees and set up a generalized least squares regression using a nonlinear model. This was successful for LACA, LAGM, and PIOB, but not for PISI and PISY due to small sample sizes, where current values of PIOB are used as a first estimate (Table 2. For each tree in the simulation, the maximum actual growth can be estimated with the following equation.

$$TRW_{Species\ j, year\ t} = exp(gdbasalconst + gdbasalfac * t + gdbasalfacq * t^2) \tag{2}$$

where TRW is the tree-ring width for species *i* at one year depending on the fitted parameters gdbasalconst, gdbasalfac, and gdbasalfacq.

Biomass data were prepared following the protocol of Shevtsova et al. (2020) and allometric relationships were established to empirically estimate the leaf area (LA) from total leaf biomass for each tree (Eq. 3), followed by a log-log linear regression forced to pass through the origin employing the basal diameter as explanatory variable (Eq. 4). To estimate the LA for each tree, we used specific leaf area (SLA) parameters to translate from the dry weight of needles to leaf area (Eq. 3). For each species, the SLA was extracted from literature values: $SLA_{LAGM} = 120$ cm² g$^{-1}$ (Xian-kui et al., 2015), which was also used for the closely related sister species LACA, $SLA_{PIOB} = 50$ cm² g$^{-1}$ (Konôpková et al., 2020), $SLA_{PISY} = 50$ cm² g$^{-1}$ (extracted from the most recent source Błasiak et al., 2021, although other values are reported, 34 cm² g$^{-1}$ in Reich et al., 1998, 40 cm² g$^{-1}$ in Mencuccini & Bonosi, 2001). For PISI no source for SLA values was found and we assume it is similar to PISY.

$$LA_{Tree\ i, Species\ j} = BM_{dry\ needle} SLA_{Species\ j}/100 \tag{3}$$

where *BM* is the biomass of tree *i* in g, and *SLA* is the specific leaf area for species *j*.

$$\log(LA_{Tree\ i,Species\ j}) = a * \log(DB_{Tree\ i}) + 0 \tag{4}$$

where *a* is the slope of the linear model fit and *DB* is the basal diameter of the tree *i.* for species *j*

During simulations runs with LAVESI, the LA for each individual tree is estimated based on the fitted linear regression model using the following equation.

$$\widehat{LA_{i,j}} = exp(a * log(DB_{Tree\ i})) \tag{5}$$

The leaf area index (LAI) of each CryoGrid 10x10 m grid cell in LAVESI is then the sum of leaf area values of present trees. When a tree crown area covers more than one cell, the value is distance-weighted on the closest grid cells. For each species, the crown radius is estimated from field data with a log-log linear regression and the slope and y-intercept are used in LAVESI, parameters *crownradiusestsslope* and *crownradiusestinterc*, respectively (Table 2).

### 2.1.4 Addition of a dynamic litter layer and estimating active-layer thickness (ALT)

A dynamic, growing litter layer with constant growth of 0.5 cm yr$^{-1}$ and stochastic disturbance effects was introduced in the *Environmentupdate*-function of LAVESI. When the parameter "*litterlayer*" is switched on, each of the 20 cm grid cells have a chance that the *litterlayerheight* can be reduced. This is stochastically implemented and for each year there is a 10% chance the litter layer is reduced by 10%, a 9% chance of a 25% reduction, a 0.9% chance of 50%, a 0.09% chance of 90%, and a 0.01% chance of a 99% reduction. This leads to a litter layer of ~15 cm in the areas of interest in simulation runs, as is observed in the region of interest (Kruse et al., 2019a). With this functionality, locally acting insulation effects are included in the estimation of the actual ALT in one environment grid cell. The estimation of the maximum active-layer thickness was already introduced in the original setup of LAVESI (Kruse et al., 2016) and still serves as a first estimate thus making it possible to run a stand-alone simulation of LAVESI without coupling to the CryoGrid (see below).

### 2.1.5 Model validation

We compared results from simulations until year 2015 with field inventories from the 2018 expedition and literature values, focusing on the following key regions: Lake Khamra (westernmost, warmest), Nyurba (intermediate, climate station), and Spasskaya Pad (easternmost for boreal forests of Yakutia) for which we used literature values for comparison. Values were in the range of expected results.

### 2.2 The 1D permafrost model CryoGrid

### 2.2.1 Model description

The model used to simulate the thermo-hydrological interactions between permafrost ground and the forest canopy is based on CryoGrid (originally described in Westermann et al., 2016). CryoGrid is a one-dimensional, numerical land surface model

that simulates the thermo-hydrological regime of permafrost ground by numerically solving the heat-conduction equation. The CryoGrid model has recently been extended by a multilayer canopy module developed by Bonan et al. (2014) for use in boreal permafrost regions (see Stuenzi et al., 2021a and 2021b, for model details). The multilayer canopy model provides a comprehensive parameterization of fluxes from the ground, through the canopy layer up to a roughness sublayer. In combination with CryoGrid the canopy model replaces the standard surface energy balance scheme while soil state variables are passed back to the forest module. Following Stuenzi et al. (2021b), a realistic canopy structure is simulated by allowing fractional compositions of deciduous and evergreen taxa within a simulated forest stand.

### 2.2.1.1. Purpose

The model CryoGrid-Vegetation (Stuenzi et al., 2021a) was set up to understand the heat and water exchange between the atmosphere, boreal forest and permafrost. The coupled multilayer vegetation-permafrost model reproduces the energy transfer and thermal regime of typical boreal permafrost ecosystems at different study sites in boreal permafrost regions.

### 2.2.1.2. Entities, state variables, and scales

Model entities are multiple layers of atmosphere and vegetation (based on Bonan et al., 2018), and permafrost (based on CryoGrid, Westermann et al., 2016). The physically based, numerical land surface model simulates the radiative heat and water transfer through the atmosphere, vegetation and ground at a 1D scale.

The simulation proceeds at a 5-minute time step. The numerical model simulates the above- and below-ground temperature field based on temporally changing conditions at the ground-surface and top of the canopy-atmosphere boundaries.

### 2.2.1.3. Process overview and scheduling

To simulate the thermo-hydrological regime of the permafrost ground CryoGrid solves the one-dimensional heat equation numerically including groundwater phase change. The canopy model was coupled to CryoGrid by replacing its standard surface energy balance scheme while soil state variables are passed back to the forest module. The vegetation module forms the upper boundary layer of the model and provides a comprehensive parameterization of fluxes from the ground, through the canopy up to the roughness sublayer. This allows the simulation of diverse forest canopy structures and their impact on the vertical moisture and energy transfer. The exchange of radiation, sensible heat, condensation, and evaporation at the different layers are simulated with a surface energy balance scheme based on atmospheric stability functions. In every time step top of the canopy incoming radiation and precipitation are partitioned at each layer throughout the canopy. The change of internal energy in the subsurface domain (ground) over time is composed of fluxes across the upper (surface energy balance below the canopy) and lower (geothermal heat flux at 100 m depth, $0.05 W/m^2$) boundaries. The model simulates the evolution of the snow cover based on an extensive CROCUS-based snowpack scheme (Zweigel et al., 2021). Furthermore, rain- and snowfall are intercepted throughout the canopy with only a fraction reaching the ground directly as throughfall. The remaining water/snow is added to the canopy layers as canopy water/snow, which either evaporates or reaches the ground as canopy drip

or stem flow in the following time steps. The model is forced by standard meteorological variables, which can be obtained from automatic weather stations, reanalysis products, or climate models. The required forcing data include air temperature, precipitation (solid and liquid), wind speed, incoming short- and longwave radiation, humidity, and air pressure (Westermann et al., 2016).

**2.2.2 Model validation and parameters**

This entire model setup has previously been extensively validated for different study sites throughout our study region, including Nyurba (63.08°N, 117.99°E), Spasskaya Pad (62.14°N, 129.37°E), and Ilirney (67.40°N, 168.37°E) (Stuenzi et al., 2021a and 2021b). Validation exercises were carried out based on measured and modelled ground surface temperature (GST), active-layer thickness (ALT), soil moisture, Bowen ratio, and short- and longwave radiation below and above the canopy. Parameters defining the canopy, snow, and soil properties were set according to literature values, model documentation, and own measurements (see Stuenzi et al., 2021b for constants and multilayer canopy parameter choices). Table 3 summarizes the parameter choices for the three different sites. Table C1 summarizes the commonly used CryoGrid parameters.

**2.3 Coupling the models**

The coupled model set-up benefits from the detailed process implementation gained while developing the individual models and brings the 1D to a landscape simulation. Therefore, we can reproduce the energy transfer and thermal regime in permafrost ground as well as the radiation budget, nitrogen and photosynthetic profiles, canopy turbulence, and leaf fluxes, while at the same time predicting the expected establishment, die-off, and treeline movements of larch forests (Fig. 2). In our analyses, we focus on vegetation and permafrost dynamics and reveal the magnitudes of different feedback processes between permafrost, vegetation, and current and future climate in Siberia.

LAVESI serves as host model and can now be set to call individual CryoGrid instances in a given year. For this, the data in LAVESI are aggregated on a 10x10 m grid superimposed on the 20x20 cm grid. Key state variables are leaf area index (LAI), stem area index (SAI), fraction of deciduous species, litter layer height, organic layer height, albedo, and the soil moisture in percent (= plant available water, PAW), which are provided to CryoGrid. These values can either be sorted by LAI and exported for 5 quartiles (implemented but not used here) or from the three areas that are equal slices from left to right (used here, see Appendix B1-B3). When the output file is created, LAVESI can be set to either start CryoGrid directly via a system call or scheduling the instance with a bash file for the workload manager slurm (Yoo et al., 2003). Based on the key state variables provided by LAVESI for each of the areas, CryoGrid starts three (or five) parallel simulations. Once the output has been written, LAVESI reads the file and produces for the three levels anomalies for available soil water and active-layer thickness. With these anomalies, the 10x10 m CryoGrid-grid in LAVESI is filled and from this, the anomalies used to calculate the new values for each 20x20 cm environment grid cell. When the quartile-mode is set, the state values are assigned to this grid calculated by linear interpolation of the LAI-sorted state values and anomalies are calculated as in the other mode.

The multilayer canopy model in CryoGrid requires a minimum LAI of 0.7 $m^2$ $m^{-2}$ and a minimum height of 1 m to successfully build the radiative transfer scheme from the atmosphere to the ground, therefore forest covers below these values are ignored.

## 2.4 Forcing data and landscape of focus areas

The meteorological forcing data required by the multilayer canopy-permafrost model (air temperature, air pressure, wind speed, relative humidity, solid and liquid precipitation, incoming long- and shortwave radiation, and cloud cover) are obtained from ERA-5 (ECMWF Reanalysis, Hersbach et al., 2018). The data are extracted for the focus regions Nyurba 63.08°N, 117.99°E (covering sites EN18067,-68,-70), Spasskaya Pad 62.14°N, 129.37°E, and Lake Khamra 59.97°N, 112.96°E (covering EN18079–83; Fig. 1).

To provide a millennia-long time series for model spin-up of LAVESI these series were matched to historical climate data for the forcing retrieved from the 0.5°x0.5° Climate Research Unit gridded Time Series (CRU TS version 3.23) monthly data (1901–2014) (Harris et al., 2020). By repeating the 20th century data in a loop, a 2100-year long monthly climate series was established from 1 to 2100 CE for each focus region using the RCP 2.6 prediction scenario.

## 2.5 Simulation experiments

We forced LAVESI simulations with the RCP 2.6 climate scenario calling CryoGrid first in 2015 and yearly in the following years, letting the simulation run until the year 2030. Simulation runs were started with the updated LAVESI version on an empty landscape with true topography starting at 1 CE to allow for spin up and ending in 2100 CE. Into the empty landscape, seeds (5000 $ha^{-1}$ $yr^{-1}$) for initiating population establishment were introduced for the first 50 years. Subsequently, only 100 $ha^{-1}$ $yr^{-1}$ seeds were introduced to allow for re-establishment after a complete die-out of trees on the whole simulation area. Each simulation was rerun without calling CryoGrid to compare the differences when the improved active-layer thickness and available soil water is used.

In addition to the simulation that uses equal proportions of seeds of each species introduced into the simulation area, we started individual simulations for each single species.

## 2.6 Statistical analyses

All statistical analyses in this study were performed in R 3.6.1 (R Core Team, 2019), mostly using included standard functions, with the addition of functions from the package "lattice" (Sarkar, 2008) for plotting the data.

# 3 Results

## 3.1 Simulations results with LAVESI

A gradient of population densities (expressed in LAI values) forms, which negatively follow the TWI gradient on all sites (I: left, driest to III: right, wettest, Fig. 5 & 6 & 7). Further, an increase of larch dominance towards nearly pure larch tree stands can be observed from Khamra (southwest, warmest) via Nyurba (intermediate) to Spasskaya Pad (northeast, coldest). Stand densities are highest in the mixed-species simulations at Nyurba (~1.9 m²/m²), larger than at warmers site Khamra (~1.5 m²/m²) and lowest at Spasskaya (0.9 m²/m²). Regarding other species, PISY is present in mixed stands in small numbers and grows only in open stands in simulations with only this species, suggesting that this species prefers a certain environment (Appendix B & D). PIOB performs better in single mono-specific runs leading to larger LAI than in mixed stands, but also reaches dense populations in warm areas (Khamra) and has smallest sizes at coldest sites (Spasskaya Pad, Fig. 5 & 6 & 7). LAGM grows under most conditions but not in the wettest areas (highest TWI values, Appendix D).

## 3.2 Comparing simulations with LAVESI and the coupled version

The values are very similar for the runs with all and individual species. In nearly all years, LAVESI's ALT values are higher by up to 20 cm (mean over all is 109.6±11.4 cm versus 96.1±10.2 cm, which is ~14.1%) at all focus regions (Fig. 3). The soil moisture anomaly fluctuates around 0% at Lake Khamra, is lower in the coupled model for Nyurba by ~10%, and Spasskaya Pad by ~20% than in simulations using only LAVESI (Fig. 4).

Smaller population sizes can be observed in all simulations leading to a drop in LAI values when LAVESI is updated by CryoGrid (comparing lower to upper panels in Fig. 5 & 6 & 7). In CryoGrid coupled runs, species grow with less dense stands but still cover the same area. In the coupled runs, populations die out in some cases at the end of the simulation at year 2030 (Appendix D).

# 4 Discussion

## 4.1 Simulation performance

Species preference matches observations and expectations (Table 4, Kuznetsova et al., 2010). Larches have a wide ecological niche and are widespread (Mamet et al., 2019). They are generalists and best adapted to the harsh Siberian environments that were predominantly wet but are now become drier with global warming (Churakova et al., 2021, Kharuk et al., 2021). *Picea obovata* grows best in the westernmost, warm areas and reaches larger LAI/biomass than when growing in mixed stands competing with other species (Kharuk et al., 2007). This is as expected and competition, which, as Wieczorek et al. (2017) shows, seems to be a strong factor dampening the response of tree stands when climatic conditions improve. Further, the simulation of denser stands at the Khamra site contradicts the general observation that mixed-species stands are more

productive/denser as the niches are occupied (Liang et al., 2016), but depending on the stand structure it could be negative (Zeller et al., 2018) and is in line with the observation of Chen et al. (2003).

## 4.2 The coupled version LAVESI-CryoGrid

The simulations using site information for the three focus sites yield dense tree stands in LAVESI simulations but not in the coupled version. The coupled model results in smaller key soil parameters, active-layer thickness (ALT), and plant available water (PAW). PAW has a strong impact as trees grow poorly in conditions exceeding drought and waterlogging thresholds (e.g. Liang et al., 2014, Mamet et al., 2019, Lawrence et al., 2015, Barber et al., 2000). Drought leads to a higher mortality of trees and in consequence, population simulations are driven close to extinction within the simulation duration of 15 years. However, when drought-adapted *Pinus sylvestris* occupies the niche, there is nearly no change and it could, in the end, colonize the simulation areas. This matches expectations and implies further that the model is reproducing the natural dynamics well (Table 4).

As fires become more intense and frequent under global warming, spruce or other species may become dominant rather than shade-intolerant larch species. In the currently naturally deciduous, larch covered areas, evergreen taxa may invade and change the heat fluxes and energy balance with the threat of entering a positive feedback loop such as a deepening of the ALT (Bonan, 2008, Stocker et al., 2013, Stuenzi et al., 2021b).

In general, technical issues arise when coupling models and implementing I/O indirectly via output files. The two different time-steps (years in LAVESI vs. 5 minutes in CryoGrid) and computational speeds lead to long computation times and high requirements of computational resources of the coupled version. To avoid any delay, a parallelization of CryoGrid simulations, as implemented here, is highly recommended, especially for dry study sites where a simulated year in CryoGrid can take up to 4 hours. We find that simulations of homogeneous areas perform best and especially that the exchange is set up using three to five instances sorted by LAI. The constraint lies here and more instances would improve the representation of variants of deciduous/evergreen covered plots but these improved LAVESI simulations come at the cost of computation time when not using the parallelized version as developed for this study.

## 5 Conclusions

The as-is application of LAVESI overestimates ALT values by around 14% therefore we advise using the implemented correction from CryoGrid for forecasting forest dynamics in the proposed coupled version. The 3D simulations provide a way to understand permafrost distribution and interactions with vegetation. Further implementations, tracked online in the github repository of LAVESI, include spatially explicit fire and trait variation and adaptation. This public sharing of the source code plus advances in both models allows the easy exchange, development, and adaptation to further regions. This and the simple set-up make the coupled model version easy to implement and thus offers a wide applicability. However, fieldwork or literature

values about present species and stand structure as well as soil moisture and temperature time series from remote areas are necessary to adjust parameters and adapt the model and species to local site conditions, which is an issue for the vast remote areas in Siberia. With increasing data from these remote areas, such as better satellite imagery coverage and resolution, the collection of more detailed field data (loggers recording soil temperatures and moisture in the active layer), and monitoring of

5 permafrost dynamics and tree growth, the drivers of forest dynamics may be disentangled and thus improve the model.

## 6 Acknowledgements

We are grateful for the support of Sven N Willner in parallelizing LAVESI. The data and experience gained on the successful Expedition Chukotka and Central Yakutia in 2018 was the basis for the developments and we thank especially Luidmila A. Pestryakova for her support and leadership of the expedition, but all participants involved in the expedition. We thank Luca

Farcas and Christopher Guth for their support in preparing samples and establishing tree-ring chronologies, and Ingo Heinrich from the GFZ dendro lab for their support of the dendroecological analyses. We thank Iuliia Shevtsova for supporting the organization and analyses of the biomass data. We thank especially the HPC support at AWI mainly Natalya Rakowski and Malte Thoma for their assistant in the simulation setup. We would like to thank Cathy Jenks as well as two anonymous reviewers for proofreading and their comments that led to an improved version of the manuscript.

This study has been supported by the ERC consolidator grant Glacial Legacy to Ulrike Herzschuh (no. 772852). Further, the work was supported by the Federal Ministry of Education and Research (BMBF) of Germany through a grant to Moritz Langer (no. 01LN1709A). Funding was additionally provided by the Helmholtz Association in the framework of MOSES (Modular Observation Solutions for Earth Systems).

## 7 Data availability

The CryoGrid code is available on Zenodo (https://doi.org/10.5281/zenodo.5119987). LAVESI is publicly available on GitHub at https://github.com/StefanKruse/LAVESI the branch used for this study is CryoGrid_multispecies (https://github.com/StefanKruse/LAVESI/tree/CryoGrid_multispecies) and the commit used for the simulations for this study is 93a9767 tagged as LAVESI-CryoGrid v1.0 (https://github.com/StefanKruse/LAVESI/releases/tag/1.0) and the final commit will be permanently stored on Zenodo upon acceptance of the manuscript.

Biomass data and tree-ring growth information for the species present at the ROI is available upon request and will be stored publicly upon acceptance of the manuscript to PANGAEA https://www.pangaea.de/

## Author contributions

Stefan Kruse SK and Ulrike Herzschuh UH initiated the idea of coupling LAVESI with CryoGrid, which was jointly developed by SK, Simone M. Stuenzi SMS, and Moritz Langer ML. SK and SMS set up the coupled version and maintained the simulation

runs together. Josias Gloy JG developed the upgrade to implement multiple species efficiently in LAVESI, SK realized the implementation of the LAVESI related programming. SK analysed the data and drafted the first version of the manuscript, which was jointly edited by all authors.

The authors declare no conflict of interest.

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

**Figures and tables**

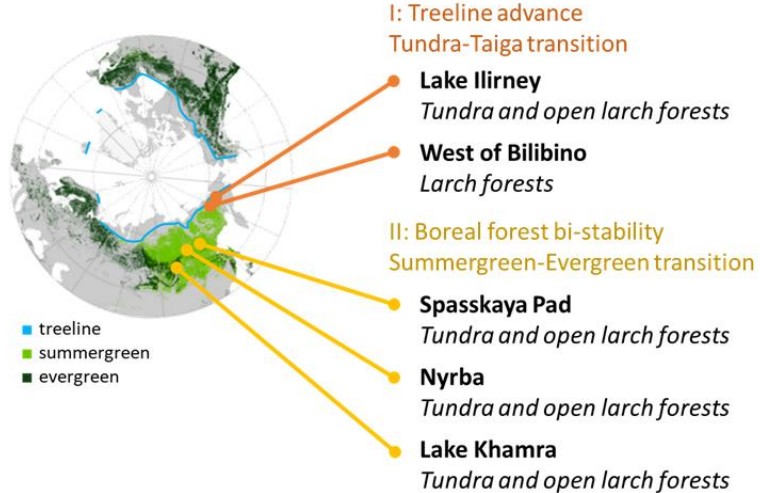

**Figure 1: Scheme of focus areas covered on the Chukotka and Central Yakutia 2018 expedition. Region II: Boreal forest bi-stability summergreen-evergreen transition needed for the expansion of species in addition to *Larix cajanderi* and *L. sibirica*.**

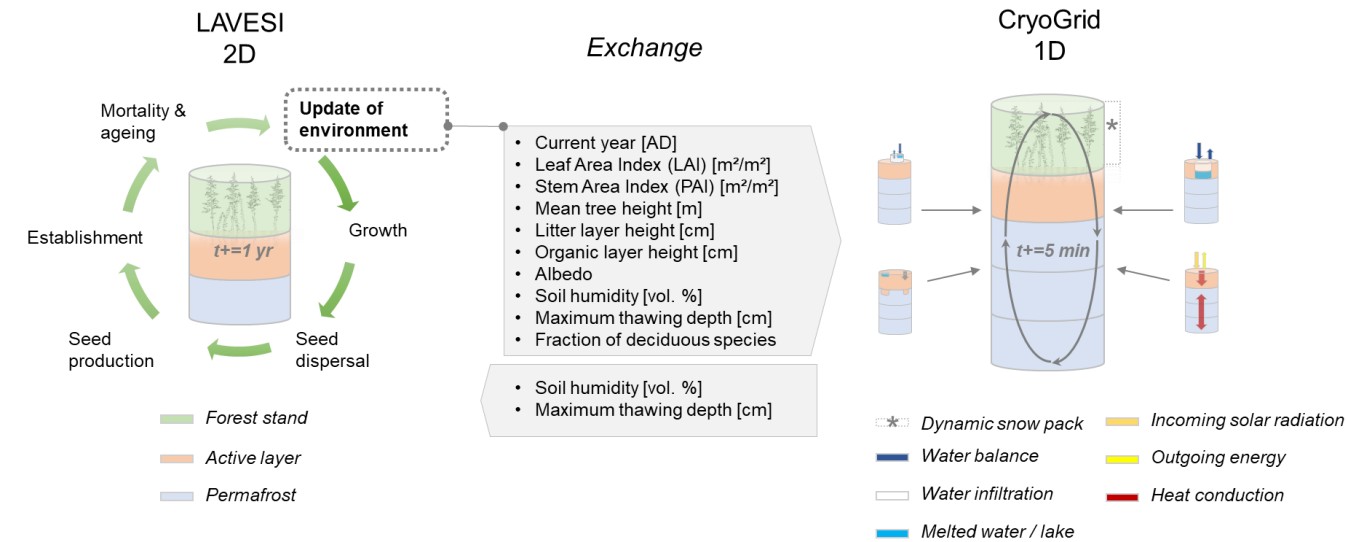

**Figure 2: Scheme of coupling CryoGrid and LAVESI and involved processes.**

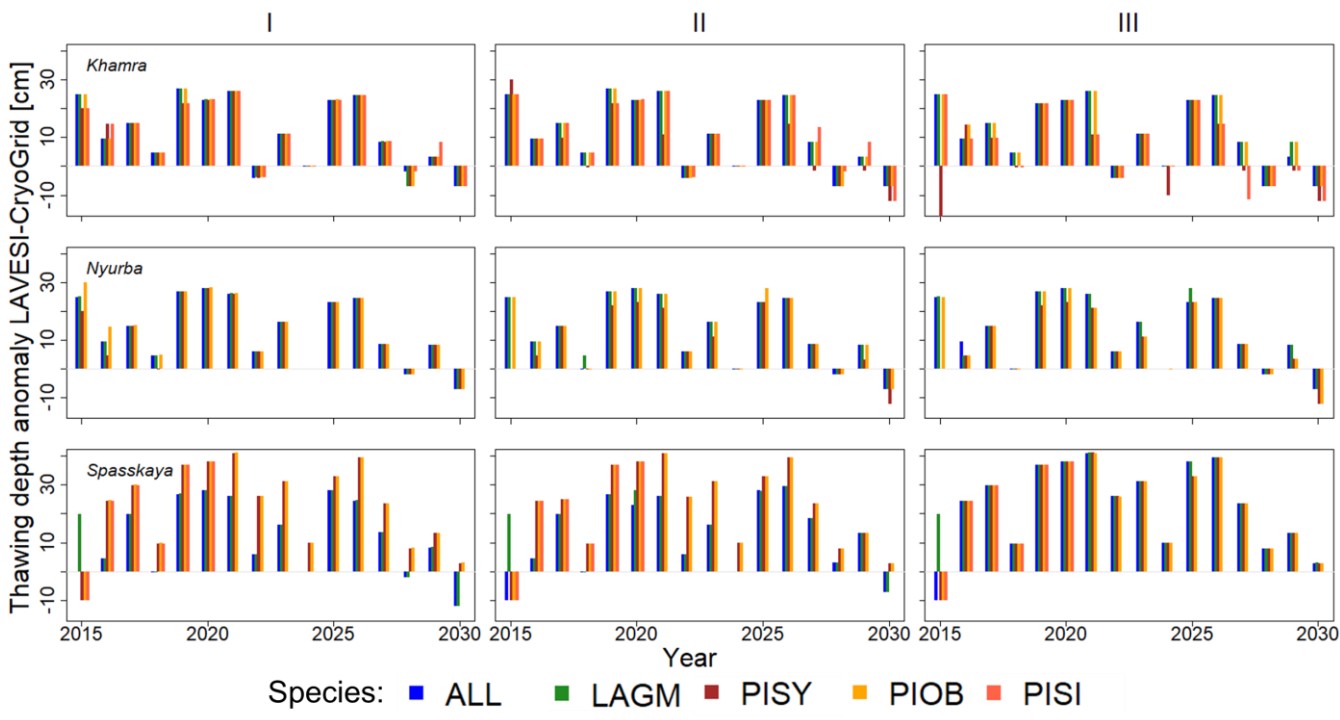

**Figure 3. Thawing depth anomaly in the coupled simulation model for all focus areas (rows) and areas within the simulation areas (columns: I, II and III) for year steps 2015-2030. LAGM *Larix gmelinii*; PIOB *Picea obovata*; PISI *Pinus sibirica*; PISY *Pinus sylvestris*.**

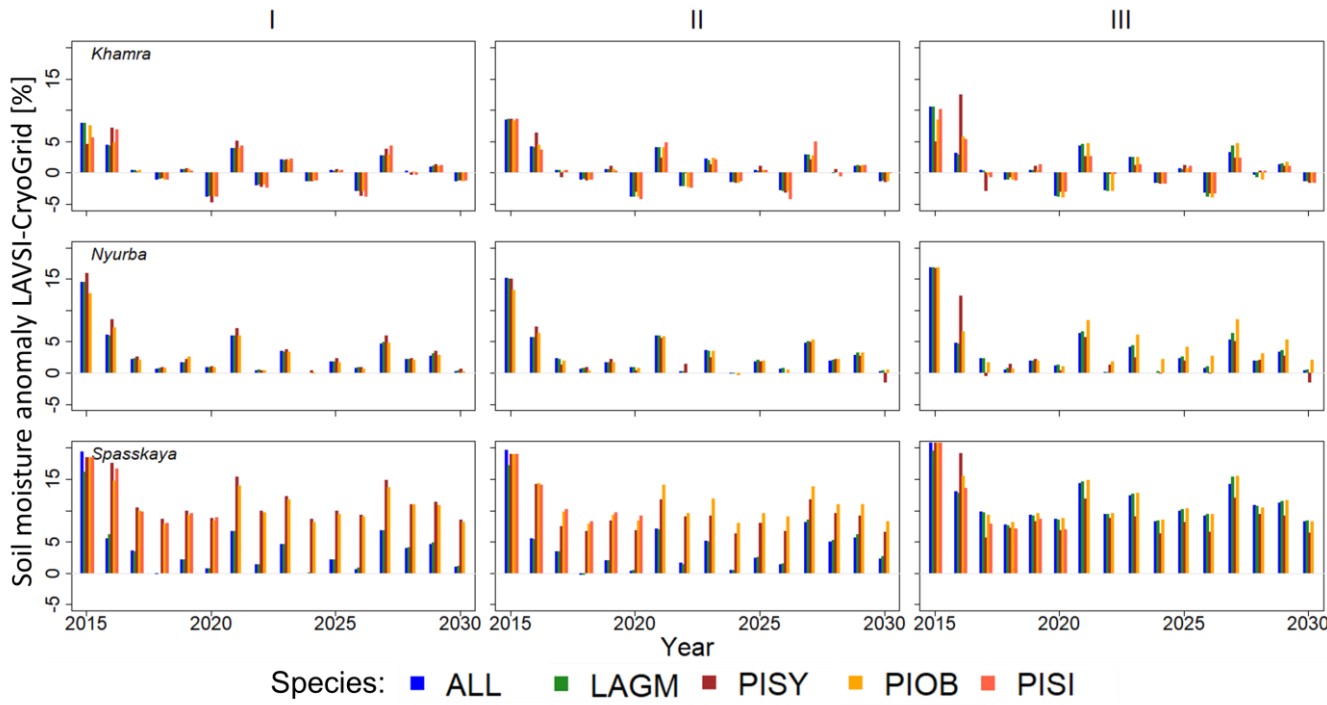

**Figure 4. Soil moisture anomalies in the coupled simulation model for all focus areas (rows) and areas within the simulation areas (columns: I, II and III) for year steps 2015-2030. LAGM** *Larix gmelinii*; **PIOB** *Picea obovata*; **PISI** *Pinus sibirica*; **PISY** *Pinus sylvestris*.

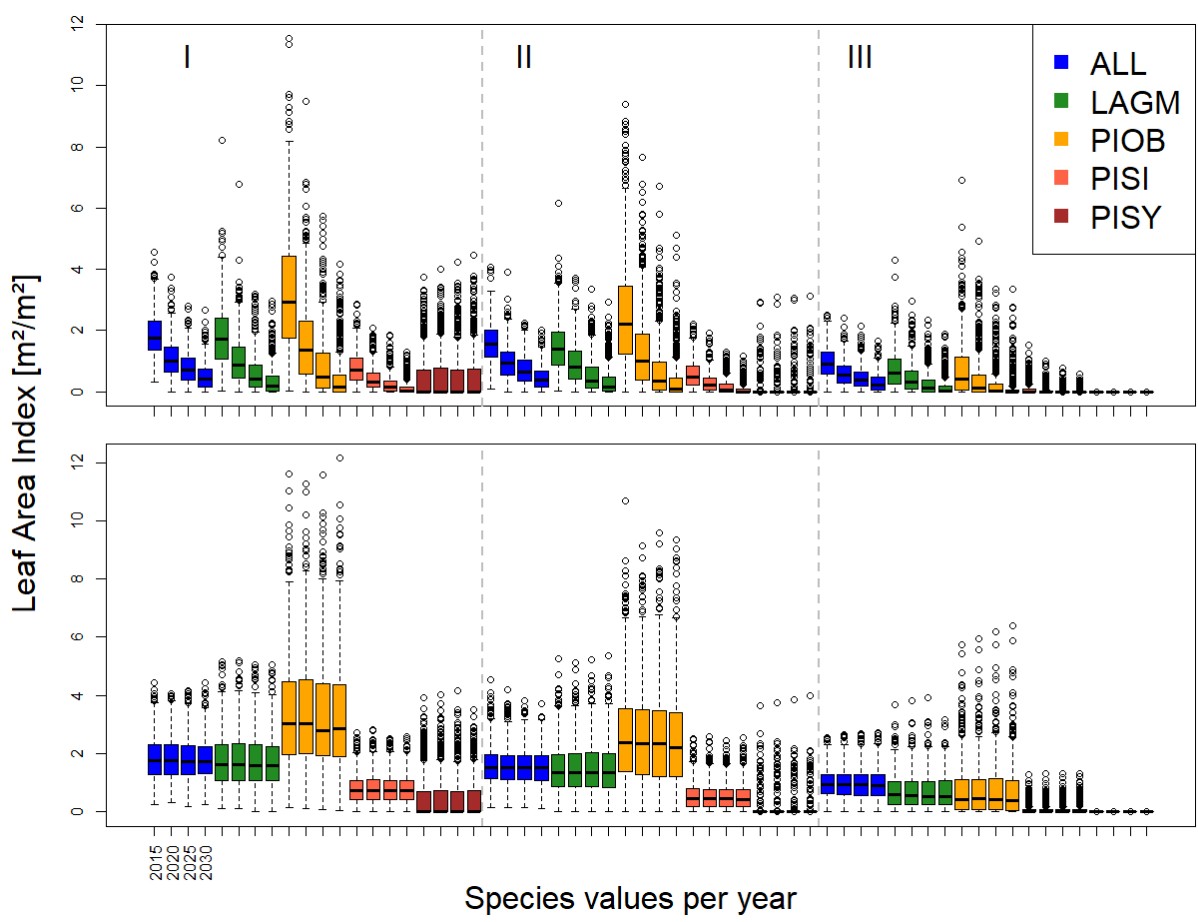

**Figure 5. Leaf area index (LAI) values at Lake Khamra for the three areas within the simulation areas (I, II, III) on the same plot at which CryoGrid was called (upper row) and only LAVESI runs (lower row). LAGM** *Larix gmelinii*; **PIOB** *Picea obovata*; **PISI** *Pinus sibirica*; **PISY** *Pinus sylvestris*.

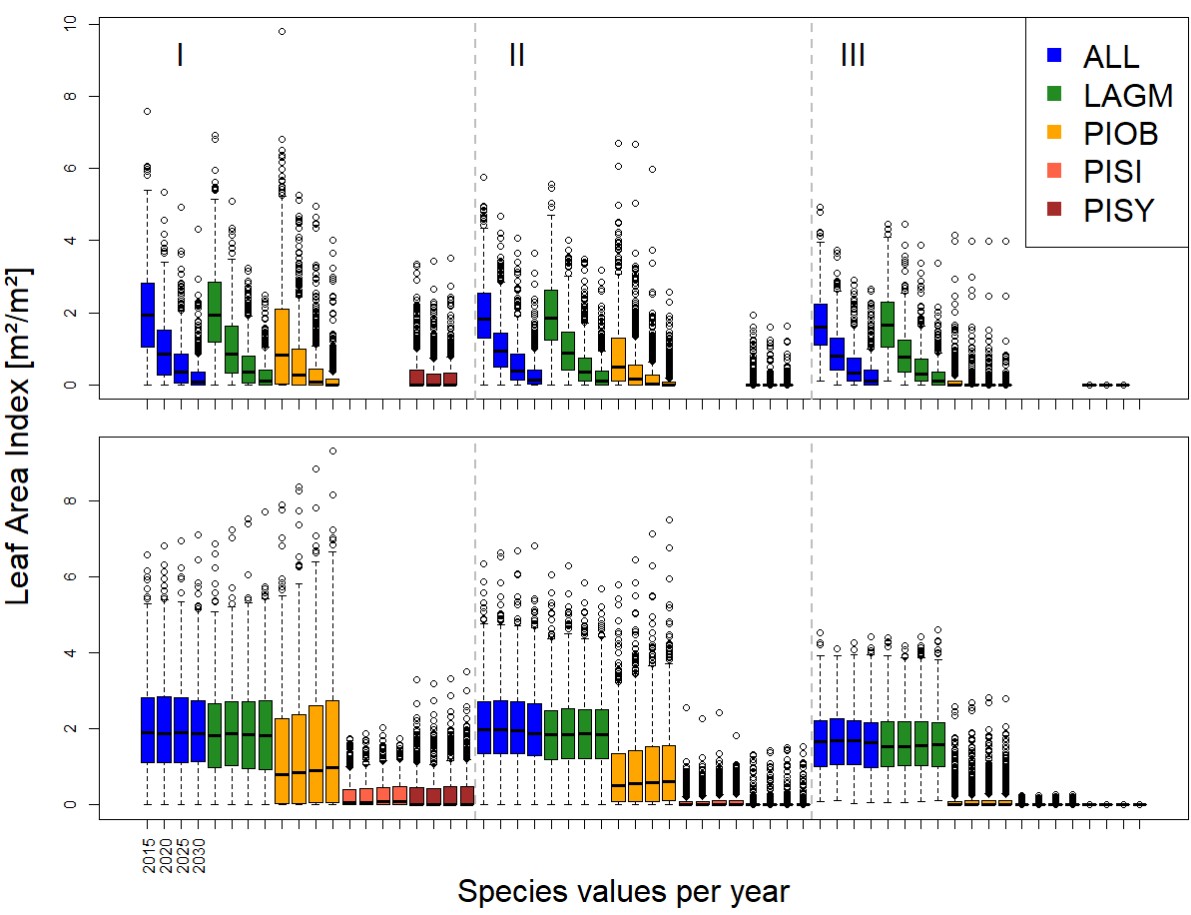

**Figure 6. Leaf area index (LAI) values at Nyurba for the three areas within the simulation areas (I, II, III) on the same plot at which CryoGrid was called (upper row) and only LAVESI runs (lower row). LAGM** *Larix gmelinii*; **PIOB** *Picea obovata*; **PISI** *Pinus sibirica*; **PISY** *Pinus sylvestris*.

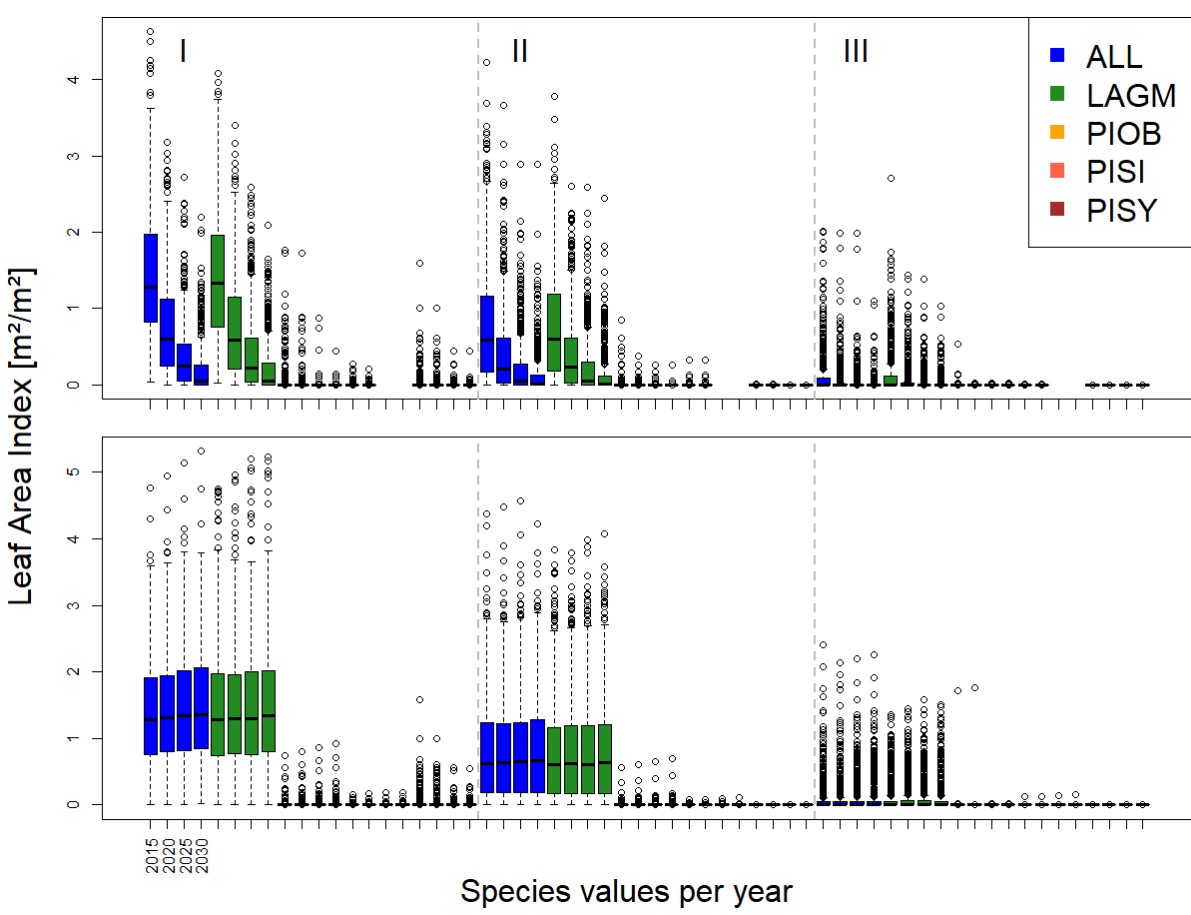

**Figure 7. Leaf area index (LAI) values at Spasskaya Pad for the three areas within the simulation areas (I, II, III) on the same plot at which CryoGrid was called (upper row) and only LAVESI runs (lower row). LAGM** *Larix gmelinii*; **PIOB** *Picea obovata*; **PISI** *Pinus sibirica*; **PISY** *Pinus sylvestris*.

**Table 1. State variables of the model with dimensions and update rates ordered by the model's hierarchy levels. Based on the table provided in Kruse et al., 2016.**

| Hierarchy | Dimension | Update rate[s] | LAVESI | LAVESI-WIND | LAVESI-CryoGrid |
|---|---|---|---|---|---|
| *Level 1 (environment)* | | | | | |
| *Biotic environment* | | | | | |
| Tree density | $(0.2 \times 0.2 \text{ m})^{-1}$ | yearly | X | X | X |
| Litter layer height | cm | yearly | | | X |
| Leaf area index LAI of all species | m²/m² | yearly | | | X |
| Leaf area index LAI of deciduous species | m²/m² | yearly | | | X |
| Stem area index SAI | m²/m² | yearly | | | X |
| Maximum tree height | m | yearly | | | X |
| Mean tree height | m | yearly | | | X |
| *Abiotic environment* | | | | | |
| Monthly mean temperatures | °C | * | X | X | X |
| Monthly precipitation sums | mm | * | X | X | X |
| Isotherms for January and July temperature | °C | * | X | X | X |
| 'net degree days' $NDD_0$ | - | * | X | X | X |
| 'active air temperature' $AAT_{10}$ | °C | * | X | X | X |
| Precipitation sum | mm | * | X | X | X |
| Wind speed and direction | m s$^{-1}$ and ° | * | | X | X |
| Maximum basal diameter growth and at breast height | cm | * | X | X | X |
| Maximum thaw depth | cm | yearly | | | X |
| Litter layer height | cm | yearly | | | X |
| Elevation | m | * | | | X |
| TWI | - | * | | | X |
| Slope | ° | * | | | X |
| Soil moisture | vol. % | yearly | | | X |

| *Level 2 (individuals)* | | | | | |
|---|---|---|---|---|---|
| *Trees* | | | | | |
| Position (*x,y* coordinates) | m | * | X | X | X |
| Year of establishment | year AD | * | X | X | |
| Diameter at basal and breast height | cm | yearly | X | X | X |
| Relative diameter growth at basal and breast height | - | yearly | X | X | X |
| List of diameters at breast height | cm | yearly | X | X | |
| Height | cm | yearly | X | X | X |
| Age | years | yearly | X | X | X |
| Cones (yes/no) | - | yearly | X | X | X |
| Height to bear cones | cm | after maturing | X | X | X |
| Density index | - | yearly | X | X | X |
| Species | - | * | | | X |
| Crown base | cm | yearly | | | X |
| Crown damage (relative) | % | yearly | | | X |
| *Seeds* | | | | | |
| Position (*x,y* coordinates) | m | * | X | X | X |
| Location (cone/ground) | - | * and after dispersal event | X | X | X |
| Age | year | yearly | X | X | X |
| Long dispersed seed (yes/no) | - | after dispersal event | X | X | |
| Species | - | * | | | X |

$^s$ asterisks mean updated once at establishment (trees), production (seeds) or initialization (abiotic environment); $NDD_0$ number of days exceeding 0 °C and $AAT_{10}$ sum of temperatures above 10 °C.

**Table 2. Species traits and model variable values in LAVESI either newly introduced for this study, adjusted from the initial version LAVESI v1.01 (\*) or introduced in the predecessor LAVESI-WIND v1.0 (\*\*). Values from \* Abaimov et al., 1998, or \*\* Sato et al., 2016, from own analyses, educated guess or parameter fitting.**

| Parameter | | Species | | | | | |
|---|---|---|---|---|---|---|---|
| Description | Abbreviation | *Larix gmelinii* | *Larix sibirica* | *Larix cajanderi* | *Picea obovata* | *Pinus sylvestris* | *Pinus sibirica* |
| Internal species variable [#] | *species* | 1 | 2 | 3 | 4 | 5 | 6 |
| Species abbreviation | - | LAGM | LASI | LACA | PIOB | PISY | PISI |
| Asymmetry of height estimation model | *heightloga* | 9.415 | 9.415 | 9.415 | 10.827 | 28.719 | 11.590869 |
| Centre of height estimation model | *heightlogb* | 2.83 | 2.83 | 2.83 | 3.543 | 10.939 | 4.102115 |
| Scaling factor of height estimation model | *heightlogc* | 2.214 | 2.214 | 2.214 | 2.381 | 4.916 | 3.057776 |
| Mortality rate of windthrow | *mwindthrow* | 0.01 | 0.01 | 0.01 | 0.01 | 0.01 | 0.01 |
| Minimum depth of active layer table [cm] | *minactivelayer* | 20\* | 200 | 20 | 200 | 100 | 200 |
| Minimum available soil water content [%] | *minsoilwater* | 21.1\*\* | 10 | 10 | 10 | 10 | 25 |
| Rooting depth [cm] | *rootingdepth* | 50\* | 100 | 20 | 200 | 100 | 100 |
| Relative bark thickness value | *relbarkthickness* | 2 | 2 | 2 | 1.5 | 3 | 3 |
| Chance of resprouting following wildfire | *resprouting* | 0.01 | 0 | 0.01 | 0 | 0 | 0 |
| Slope of leaf biomass estimation model | *biomassleaffaca* | 1.955683 | 1.955683 | 2.162319 | 2.482039 | 2.260794 | 2.125194 |
| Slope of woody biomass estimation model | *biomasswoodfaca* | 3.553949 | 3.553949 | 3.901602 | 3.844512 | 3.257366 | 3.541813 |
| Deciduousness (binary: 1/0, yes/no) | *deciduous* | 1 | 1 | 1 | 0 | 0 | 0 |
| Slope of crown radius estimation model | *crownradiusestslope* | 0.728231 | 0.728231 | 0.9193333 | 0.6007845 | 0.7899374 | 0.5785676 |
| Intercept of crown radius estimation model | *crownradiusestinterc* | 2.794274 | 2.794274 | 2.4618496 | 2.9118007 | 2.4135727 | 2.9459064 |
| Slope of leaf area estimation model | *leafareaslope* | 2.017164 | 2.017164 | 2.236605 | 2.242359 | 2.015382 | 1.927198 |

| Parameter | | Species | | | | | |
|---|---|---|---|---|---|---|---|
| Description | Abbreviation | *Larix gmelinii* | *Larix sibirica* | *Larix cajanderi* | *Picea obovata* | *Pinus sylvestris* | *Pinus sibirica* |
| Internal species variable [#] | *species* | 1 | 2 | 3 | 4 | 5 | 6 |
| Species abbreviation | - | LAGM | LASI | LACA | PIOB | PISY | PISI |
| Minimum age to begin to bear cones [yr] (*) | *coneage* | 15 | 15 | 15 | 15 | 15 | 15 |
| Probability of seed release from cones (*) | *seedflightrate* | 0.63931 | 0.63931 | 0.63931 | 0.63931 | 0.63931 | 0.95 |
| Horizontal seed dispersal distance at wind speed of 10 km/h [m] (**) | *seedtravelbreeze* | 60.1 | 45 | 60.1 | 15 | 30 | 30 |
| Seed descent rate [m/s] (**) | *seeddescent* | 0.86 | 0.93 | 0.86 | 1.2 | 2.4 | 2.4 |
| Factor of dispersal distance (*) | *distanceratio* | 0.16 | 0.16 | 0.16 | 0.16 | 0.16 | 0.16 |
| Factor of seed productivity (*) | *seedprodfactor* | 8 | 8 | 8 | 8 | 8 | 16 |
| Background germination rate (*) | *germinationrate* | 0.01 | 0.01 | 0.01 | 0.01 | 0.01 | 0.01 |
| Influence factor of weather on germination rate (*) | *germinationweatherinfluence* | 0.447975 | 0.447975 | 0.447975 | 0.447975 | 0.447975 | 0.447975 |
| Quadratic term of the equation for basal diameter growth rate [ln(cm)/cm²] (**) | *gdbasalfacq* | -0.000133194 | -0.0009 | -0.003 | -0.000252939 | -0.000252939 | -0.000252939 |
| Linear term of the basal diameter growth function [ln(cm)/cm] (**) | *gdbasalfac* | 0.001470654 | 0.0056 | 0.03 | 0.006578208 | 0.006578208 | 0.006578208 |
| Constant term of the basal diameter growth function [ln(cm)] (**) | *gdbasalconst* | -0.805581404 | -1.01 | -1.98 | -1.319846682 | -1.319846682 | -1.319846682 |
| Quadratic term of the equation for breast height diameter growth rate [ln(cm)/cm²] (**) | *gdbreastfacq* | -0.000133194 | -0.0009 | -0.003 | -0.000252939 | -0.000252939 | -0.000252939 |
| Linear term of the breast height diameter growth function [ln(cm)/cm] (**) | *gdbreastfac* | 0.001470654 | 0.0056 | 0.03 | 0.006578208 | 0.006578208 | 0.006578208 |
| Constant term of the breast height diameter growth function [ln(cm)] (**) | *gdbreastconst* | -0.805581404 | -1.01 | -1.98 | -1.319846682 | -1.319846682 | -1.319846682 |

| Parameter | | Species | | | | | |
|---|---|---|---|---|---|---|---|
| Description | Abbreviation | *Larix gmelinii* | *Larix sibirica* | *Larix cajanderi* | *Picea obovata* | *Pinus sylvestris* | *Pinus sibirica* |
| Internal species variable [#] | *species* | 1 | 2 | 3 | 4 | 5 | 6 |
| Species abbreviation | - | LAGM | LASI | LACA | PIOB | PISY | PISI |
| Height-diameter nonlinear function slopes (H<1.3 m & ≥1.3 m) [cm·cm$^{-1}$] (*) | *dbasalheightalloslope, dbreastheightalloslope* | 42.88 | 42.88 | 42.88 | 42.88 | 42.88 | 42.88 |
| Height-diameter nonlinear function exponent (H<1.3 m & ≥1.3 m) (*) | *dbasalheightalloexp, dbreastheightalloexp* | 1 | 1 | 1 | 1 | 1 | 1 |
| Height-diameter nonlinear function slopes (H<1.3 m) [cm·cm$^{-1}$] (*) | *dbasalheightslopenonlin* | 44.43163 | 44.43163 | 44.43163 | 44.43163 | 44.43163 | 44.43163 |
| Height-diameter nonlinear function slopes (≥1.3 m) [cm$^{0.5}$·cm$^{-0.5}$] (*) | *dbreastheightslopenonlin* | 7.02 | 7.02 | 7.02 | 7.02 | 7.02 | 7.02 |
| Background mortality rate [yr$^{-1}$] (*) | *mortbg* | 0.0001 | 0.0001 | 0.0001 | 0.0001 | 0.0001 | 0.0001 |
| Maximum tree age [yrs] (*) | *maximumage* | 609 | 500 | 500* | 250 | 250 | 250 |
| Influence factor for trees older than the age limit on tree mortality (*) | *mortage* | 8.18785 | 8.18785 | 8.18785 | 8.18785 | 8.18785 | 8.18785 |
| Tree youth influence factor on tree mortality (*) | *mortyouth* | 0.762855 | 0.762855 | 0.762855 | 0.762855 | 0.762855 | 0.4 |
| Span of tree youth mortality (*) | *mortyouthinfluenceexp* | 0.79295 | 0.79295 | 0.79295 | 0.79295 | 0.79295 | 0.79295 |
| Influence exponent on current tree growth mortality (*) | *mgrowth* | 0.5 | 0.5 | 0.5 | 0.5 | 0.5 | 0.01 |
| Density influence factor on tree mortality (*) | *mdensity* | 0.5 | 0.5 | 0.5 | 0.5 | 0.5 | 0.2 |
| Weather influence factor on tree mortality (*) | *mweather* | 0.1 | 0.1 | 0.1 | 0.1 | 0.1 | 0.1 |
| Exponent scaling the height influence (**) | *heightweathermorteinflussexp* | 0.2 | 0.2 | 0.2 | 0.2 | 0.2 | 0.2 |
| Drought influence factor on tree mortality (*) | *mdrought* | 0.237805 | 0.237805 | 0.237805 | 0.1 | 0.1 | 0.5 |
| Seed mortality rate on trees (in cones) [yr$^{-1}$] (*) | *seedconemort* | 0.44724 | 0.44724 | 0.44724 | 0.44724 | 0.44724 | 0.44724 |

| Parameter | | Species | | | | | |
|---|---|---|---|---|---|---|---|
| Description | Abbreviation | *Larix gmelinii* | *Larix sibirica* | *Larix cajanderi* | *Picea obovata* | *Pinus sylvestris* | *Pinus sibirica* |
| Internal species variable [#] | *species* | 1 | 2 | 3 | 4 | 5 | 6 |
| Species abbreviation | - | LAGM | LASI | LACA | PIOB | PISY | PISI |
| Seed mortality rate at the ground [yr$^{-1}$] (*) | *seedfloormort* | 0.55803 | 0.55803 | 0.55803 | 0.55803 | 0.55803 | 0.999 |
| Maximum age of seeds [yrs] (*) | *seedmaxage* | 4* | 1* | 1* | 1 | 1 | 1 |
| Mean temperature of the coldest month (January) at the border of the species' geographical range [°C] (*) | *janthresholdtemp* | -45 | -33 | -60 | -33 | -33 | -33 |
| Fitting factor for processing temperature of the coldest month at the border of the species' geographical range (*) | *janthresholdtempcalcvalue* | 9 | 6.6 | 9 | 6.6 | 6.6 | 6.6 |
| Function parameter a-d determining curvature of the July index calculation (**) | *weathervariablea* | 0.078 | 0.163 | 0.078 | 0.163 | 0.163 | 0.163 |
| | *weathervariableb* | 14.825 | 12.319 | 14.825 | 12.319 | 12.319 | 12.319 |
| | *weathervariablec* | 0.108 | 0.168 | 0.108 | 0.168 | 0.168 | 0.168 |
| | *weathervariabled* | 0.1771 | 0.305 | 0.1771 | 0.305 | 0.305 | 0.305 |
| Inverse of the von Mises distribution's variance ($\kappa$) (**) | *kappa* | 10 | 10 | 10 | 10 | 10 | 10 |
| Gregory's parameter $C$ [cm$^{-(1-0.5m)}$] (**) | *C* | 0.6 | 0.6 | 0.6 | 0.6 | 0.6 | 0.6 |
| Gregory's parameter $m$ (**) | *M* | 1.25 | 1.25 | 1.25 | 1.25 | 1.25 | 1.25 |
| Pollen descending velocity ($V_{d, Pollen}$) [m s$^{-1}$] (**) | *velocity* | 0.126 | 0.126 | 0.126 | 0.126 | 0.126 | 0.126 |
| Factor for the actual wind direction ($\overline{\theta}$) (**) | *phi* | 1 | 1 | 1 | 1 | 1 | 1 |

**Table 3. Study site parameters for CryoGrid.**

| Study site | Soil layer depth (Litter/Organic/Mineral) | Respective soil type | ERA-interim coordinate |
|---|---|---|---|
| Nyurba | 0/0.07/0.16 | Peat/Clay/Sand | N 63.08°, E 117.99° |
| Spasskaya Pad | 0/0.08/0.16 | Peat/Clay/Sand | N 62.14°, E 129.37° |
| Khamra | 0/0.05/0.9 | Peat/Clay/Sand | N 59.98°, E 112.96° |

**Table 4. Qualitative comparison of simulation results to expectations based on observations.**

| Pattern | Expectation | Simulation study |
|---|---|---|
| *Species presence* | Khamra:<br>• mixed-species stands, relatively equal contribution of deciduous/evergreen taxa<br>• warm living taxa (PISI) present *<br>Nyurba:<br>• Mixes-species stands with larch dominance<br>• no warm living taxa (PISI) present *<br>Spasskaya Pad:<br>• pure larch forests ** | • LAGM and PIOB dominate (LAI ~1.5 m²/m²)<br>• PISI is present in low numbers (LAI ~0.5 m²/m²)<br><br>• LAGM most dense (LAI ~1.9 m²/m²)<br>• PISI grows in low numbers (LAI ~0.2 m²/m²)<br><br>• only LAGM grows (LAI ~0.9 m²/m²) |
| *Stand densities* | • density gradient: Khamra > Nyurba > Spasskaya Pad *+***<br>• species mixtures have higher densities **** | • stand densities slightly smaller at Khamra than Nyurba, lowest at Spasskaya Pad<br>• species mixtures at Nyurba and Spasskaya are slightly denser than in mono-species simulations (2 vs. 1.9 m²/m²)<br>• mono-species PIOB stands yield higher densities at Khamra |
| *Stand distribution* | • LAGM generalist vs. PIOB and PISY prefer dryer soils.*,*** | • increased drought led populations close to extinction<br>• only mono-species PISY stands are rather constant in densities |

* Kruse et al., 2019a; **Ohta et al., 2001, Sugimoto et al., 2002; *** Mamet et al., 2019; **** Liang et al., 2016

**Appendix**

**Appendix A. Complete ODD "Overview" table for LAVESI. Improvements from left to right; grey text not changed model parts, while black colour highlight novel developments.**

| ODD protocol 'Overview' | Model: | | |
|---|---|---|---|
| | LAVESI v.1.01 (Kruse et al., 2016) | LAVESI-WIND v.1.0 (Kruse et al., 2018) | LAVESI-CryoGrid v.1.0 (this study) |
| **1. Purpose** | The model was set up to understand tree stand structure and the dynamics of *Larix gmelinii* (RUPR.) RUPR. populations growing in the Siberian treeline ecotone in response to a changing climate. | The model was set up to understand tree stand structure and the migration dynamics of *Larix gmelinii* (Rupr.) Rupr. populations growing in the Siberian treeline ecotone in response to a changing climate. | The model was set up to understand tree stand structure, migration and population dynamics of boreal forests growing between the leading edge at the Siberian treeline ecotone and the southern limit in response to a changing climate and its feedbacks with permafrost soils. |

| 2. Entities, state variables, and scales | The model consists of two hierarchical levels characterized by a set of variables (Table 1): (1) simulation areas characterized by the specific biotic and abiotic environment, and (2) individual trees and seeds. Each individual square simulation area covers 100x100 m on which seeds and trees are exactly positioned by $x,y$ coordinates. Using the basal diameter of individual trees, the plot is overlaid with a tree density grid with a resolution of 0.2x0.2 m. Of the whole simulation area, the central 20x20 m represents the investigation plot ensuring a border of 40 m to the boundaries to minimize potential boundary effects. | The model consists of two hierarchical levels characterized by a set of variables (Table 1): (1) simulation areas characterized by the specific biotic and abiotic environment, and (2) individual trees and seeds. The individual simulation areas are variable and have a size of typically 100x100 m (for parameterization and for sensitivity study 100x1000 m, with the longest side north-south oriented) on which seeds and trees are exactly positioned by $x,y$ coordinates. Using the basal diameter of individual trees, the plot is overlaid with a tree density grid with a resolution of 0.2x0.2 m. | The model consists of two hierarchical levels characterized by a set of variables (Table 1): (1) simulation areas characterized by the specific biotic and abiotic environment, and (2) individual trees and seeds. The individual simulation areas are variable and have a size of typically 510x510 m (for parameterization and simulation experiments) on which seeds and trees are exactly positioned by $x,y$ coordinates. Using the basal diameter of individual trees, the plot is overlaid with a tree density grid with a resolution of 0.2x0.2 m. |
|---|---|---|---|

| | | |
|---|---|---|
| | Simulation runs proceed in yearly time steps. We performed simulations for years 1919–2011, where robust climate series were available. Additionally, to reach stabilization of population dynamics and the forcing climate series, simulations were preceded by a stabilization period with a length of 1,000 years (for parameterization, sensitivity analysis and Taymyr treeline application) or 5,001 years (for temperature experiment). All simulations start from bare ground introducing 1,000 seeds in the first 100 years and, to allow for repopulation of simulation areas after extinction, 10 seeds are added every year to the simulation areas. | Simulation runs proceed in yearly time steps. We performed simulations for years 1934–2013, where robust climate series were available. Additionally, to reach stabilization of population dynamics and the forcing climate series, simulations were preceded by a stabilization period with a length of 1,000 years (for parameterization and sensitivity analysis). All simulations start from bare ground introducing 1,000 seeds in the first 100 years and, to allow for repopulation of simulation areas after extinction, 10 seeds are added every year to the simulation areas. | Simulation runs proceed in yearly time steps. We performed simulations for years 1–2100, prolonged by RCP prediction scenarios. Additionally, to reach stabilization of population dynamics and the forcing climate series, simulations were preceded by a stabilization period with a length of 1,000 years (for parameterization and sensitivity analysis). All simulations start from bare ground introducing 5000 ha$^{-1}$ yr$^{-1}$ seeds in the first 50 years and, to allow for repopulation of simulation areas after extinction, 100 ha$^{-1}$ yr$^{-1}$ seeds are added every year to the simulation areas. |

| 3. Process overview and scheduling | The simulation proceeds in yearly time steps from the beginning to the end of the input climate time-series following a stabilization period of 1,000 years to ensure that emerging populations reach equilibrium with the environment. In each initialization phase of each simulation run, the weather data are processed and used to estimate maximum diameter growth (at basal and breast height) for each simulation year based on 10-years mean climate auxiliary variables (see details in '2.2.2 Description of sub-models' in Kruse et al., 2016). Within the growth processes of the model, these variables are used to individually estimate the current diameter growth of trees constrained by their actual biotic environment (Design concept: Sensing). Stochasticity in the model was introduced by using random numbers generated with a pseudo random | The simulation proceeds in yearly time steps from the beginning to the end of the input climate time-series following a stabilization period of 1,000 years to ensure that emerging populations reach equilibrium with the environment. In each initialization phase of each simulation run, the weather data are processed and used to estimate maximum diameter growth (at basal and breast height) for each simulation year based on 10-years mean climate auxiliary variables (see details in '2.2.2 Description of sub-models' in Kruse et al., 2016). Within the growth processes of the model, these variables are used to individually estimate the current diameter growth of trees constrained by their actual biotic environment (Design concept: Sensing). Stochasticity in the model was introduced by using random numbers generated with a pseudo random number generator (C++-function '*rand*', using the function '*srand*' for seeding and using a runtime value with the function call 'time(0)' to allow for different results between two or more | The simulation proceeds in yearly time steps from the beginning to the end of the input climate time-series, which includes a stabilization period to ensure that emerging populations reach equilibrium with the environment. In each initialization phase of each simulation run, the weather data are processed and used to estimate maximum diameter growth (at basal and breast height) for each simulation year based on 10-years mean climate auxiliary variables (see details in '2.2.2 Description of sub-models' in Kruse et al., 2016). Within the growth processes of the model, these variables are used to individually estimate the current diameter growth of trees constrained by their actual biotic (competition) and abiotic (landscape features: elevation, TWI, slope, soil moisture, active layer depth) environment (Design concept: Sensing). Stochasticity in the model was introduced by using random numbers generated with a pseudo random number generator (mt19937_64, from the random library) to allow for different results between two or more |
|---|---|---|---|

| | | |
|---|---|---|
| number generator (C++-function '*rand*', using the function '*srand*' for seeding and using a runtime value with the function call 'time(0)' to allow for different results between two or more consecutive runs of the model; Design Concept: Stochasticity). Within one simulation year, the following processes become consecutively invoked (see Fig. 2 in Kruse et al. (2016), and for detailed explanations for each process can be found in a corresponding section in '2.2.2 Description of sub-models'): **Update of environment:** Interactions between neighbouring trees are local and indirect. Basal diameters of each individual tree are used to evaluate the competition strength. We use a yearly updated density map to pass information about competition for resources between trees. (Design Concept: Interaction). **Growth:** The individual growth of basal | consecutive runs of the model; Design Concept: Stochasticity). Within one simulation year, the following processes become consecutively invoked (see Fig. 2 in Kruse et al. (2016), and for detailed explanations for each process can be found in a corresponding section in '2.2.2 Description of sub-models'): **Update of environment:** Interactions between neighbouring trees are local and indirect. Basal diameters of each individual tree are used to evaluate the competition strength. We use a yearly updated density map to pass information about competition for resources between trees. (Design Concept: Interaction). **Growth:** The individual growth of basal diameter and, if a tree reached a height of 1.3 m, of breast height diameter, is calculated from the maximum possible growth in the current year affected by the tree's density index. From the resulting diameters, the tree height is estimated differently for the two height classes, smaller and greater than 1.3 m. (Design Concept: Collectives). **Seed dispersal:** | consecutive runs of the model; Design Concept: Stochasticity). Within one simulation year, the following processes become consecutively invoked (see Fig. 2 in Kruse et al. (2016), and for detailed explanations for each process can be found in a corresponding section in '2.2.2 Description of sub-models'): **Update of environment:** Interactions between neighbouring trees are local and indirect. Basal diameters of each individual tree are used to evaluate the competition strength. We use a yearly updated density map to pass information about competition for resources between trees. (Design Concept: Interaction). **Further, a litter layer and the state variables of each grid cell are updated as well.** **Growth:** The individual growth of basal diameter and, if a tree reached a height of 1.3 m, of breast height diameter, is calculated from the maximum possible growth in the current year affected by the tree's density index **and its abiotic environment**. From the resulting diameters, the tree height is estimated differently for the two height classes, smaller and |

| | | |
|---|---|---|
| diameter and, if a tree reached a height of 1.3 m, of breast height diameter, is calculated from the maximum possible growth in the current year affected by the tree's density index. From the resulting diameters, the tree height is estimated differently for the two height classes, smaller and greater than 1.3 m. (Design Concept: Collectives). **Seed dispersal:** Seeds in 'cones' are dispersed from the parent trees, at a set rate. The dispersal directions are randomly determined with decreasing probabilities for long distances and, if dispersed seeds leave the extent of the simulated plot, they are removed from the system. **Seed production:** Trees produce seeds after the year at which they reached their stochastically estimated maturation height. The total amount depends on weather, competition, and tree size. **Establishment:** The seeds | Seeds in 'cones' are dispersed from the parent trees, at a set rate. The dispersal directions and distances are randomly determined from a ballistic flight influenced by wind speed and direction with decreasing probabilities for long distances and, if dispersed seeds leave the extent of the simulated plot, they can be introduced from the other side or only on the east-west margins, depending on the user's choice. **Seed production:** Trees produce seeds after the year at which they reached their stochastically estimated maturation height. The total amount depends on weather, competition, and tree size. Optionally, the pollen donor for the pollination of ovules of seeds produced can be selected by a wind-determined and distance-dependent probability distribution function using a von Mises distribution. **Establishment:** The seeds that lie on the ground germinate at a rate depending on current weather conditions. **Mortality:** Individual trees or seeds die, i.e. they become removed from the plot, at a specified mortality rate. For trees | greater than 1.3 m. (Design Concept: Collectives). **Seed dispersal:** Seeds in 'cones' are dispersed from the parent trees, at a set rate. The dispersal directions and distances are randomly determined from a ballistic flight influenced by wind speed and direction with decreasing probabilities for long distances **and only to places lower than the release height. I**f dispersed seeds leave the extent of the simulated plot they are removed from the system, but optionally they could be introduced from the other side or only on the east-west margins, depending on the user's choice. **Seed production:** Trees produce seeds after the year at which they reached their stochastically estimated maturation height. The total amount depends on weather, competition, and tree size. Optionally, the pollen donor for the pollination of ovules of seeds produced can be selected by a wind-determined and distance-dependent probability distribution function using a von Mises distribution. **Establishment:** The seeds that lie on the ground germinate at a rate depending on |

| | | |
|---|---|---|
| that lie on the ground germinate at a rate depending on current weather conditions. **Mortality:** Individual trees or seeds die, i.e. they become removed from the plot, at a specified mortality rate. For trees this is deduced from long-term mean weather values, a drought index, surrounding tree density, tree age and size, plus a background mortality rate. Seeds on the other hand have the same constant mortality rate whether on trees and or the ground. (Design Concept: Emergence). **Ageing:** Finally, the age of seeds and trees increases once a year and seeds are removed from the system when they reach a defined species age limit. | this is deduced from long-term mean weather values, a drought index, surrounding tree density, tree age and size, plus a background mortality rate. Seeds on the other hand have the same constant mortality rate whether on trees and or the ground. (Design Concept: Emergence). **Ageing:** Finally, the age of seeds and trees increases once a year and seeds are removed from the system when they reach a defined species age limit. | current weather conditions and is constrained by the actual litter layer height. **Mortality:** Individual trees or seeds die, i.e. they become removed from the plot, at a specified mortality rate. For trees this is deduced from long-term mean weather values, a drought index, surrounding tree density, tree age and size, plus a background mortality rate. Seeds on the other hand have the same constant mortality rate whether on trees and or the ground. (Design Concept: Emergence). **Ageing:** Finally, the age of seeds and trees increases once a year and seeds are removed from the system when they reach a defined species age limit. |

**Appendix B. Landscape defining the focus region's plot area.**

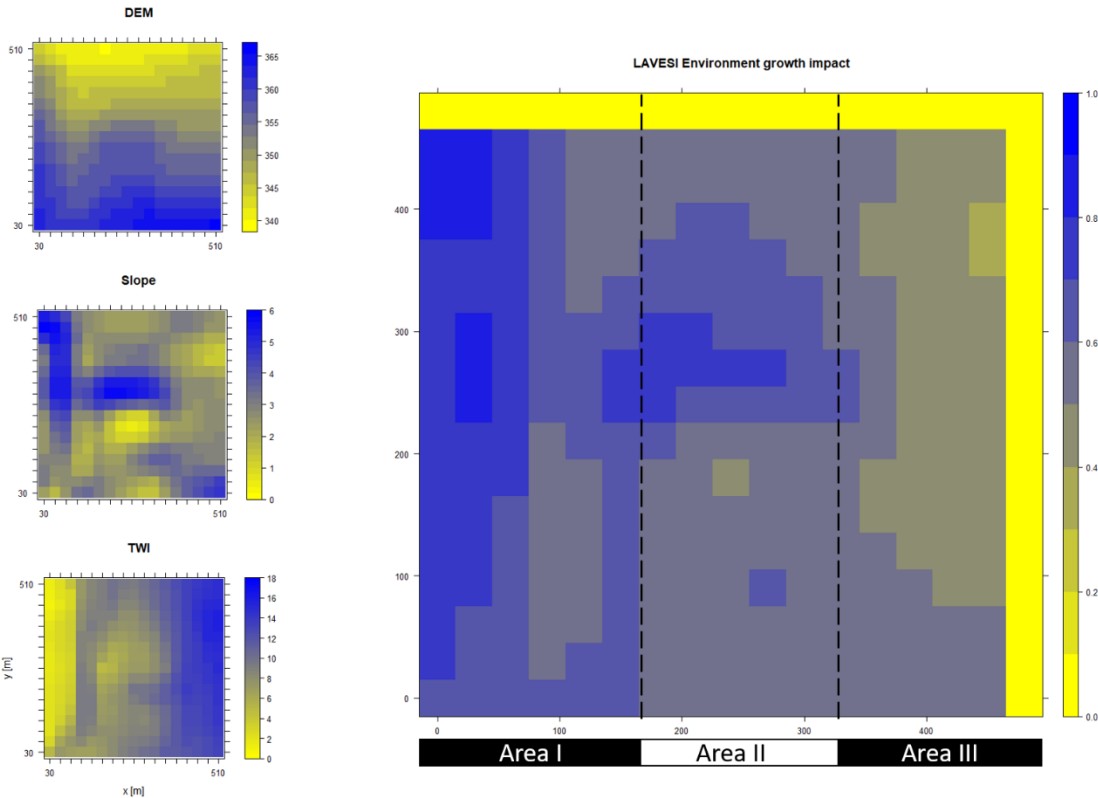

**Figure B1. Elevation (DEM), slope angle, and terrain water index (TWI) define the environment growth impact (0 no growth possible; 1 good, no constraints) using an empirically fitted function for present forest growth at area of interest Khamra.**

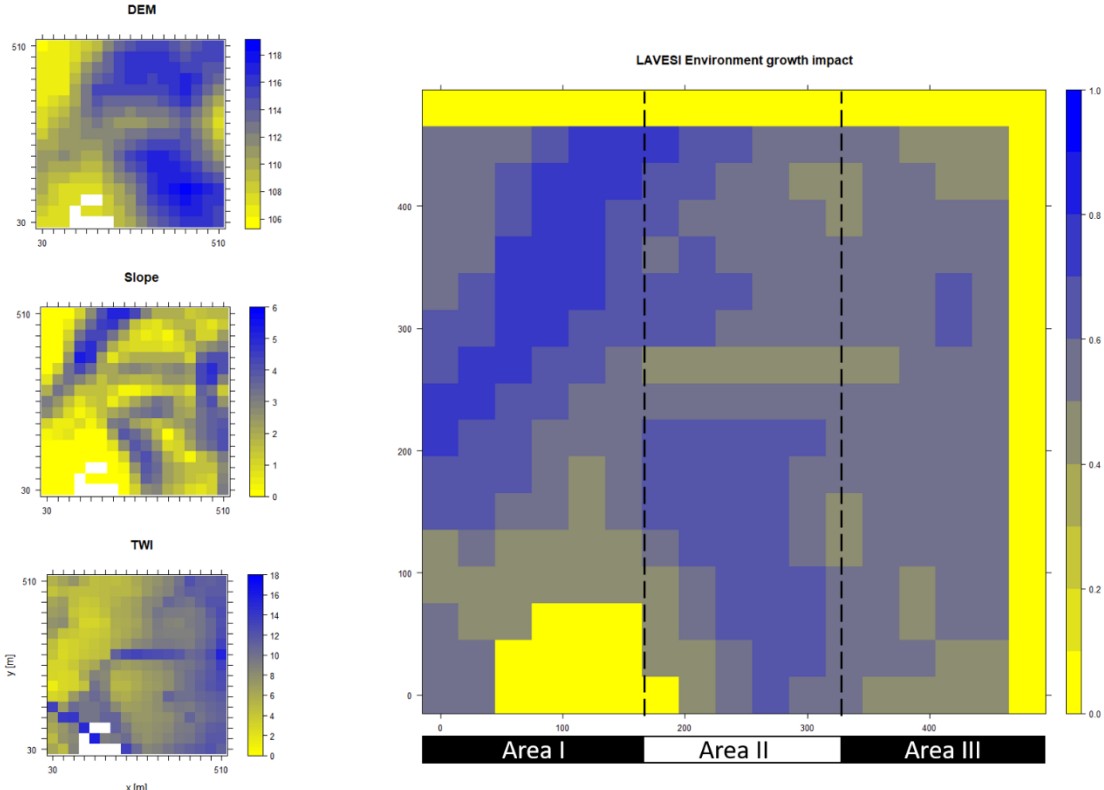

**Figure B2.** Elevation (DEM), slope angle, and terrain water index (TWI) define the environment growth impact (0 no growth possible; 1 good, no constraints) using an empirically fitted function for present forest growth at area of interest Nyurba.

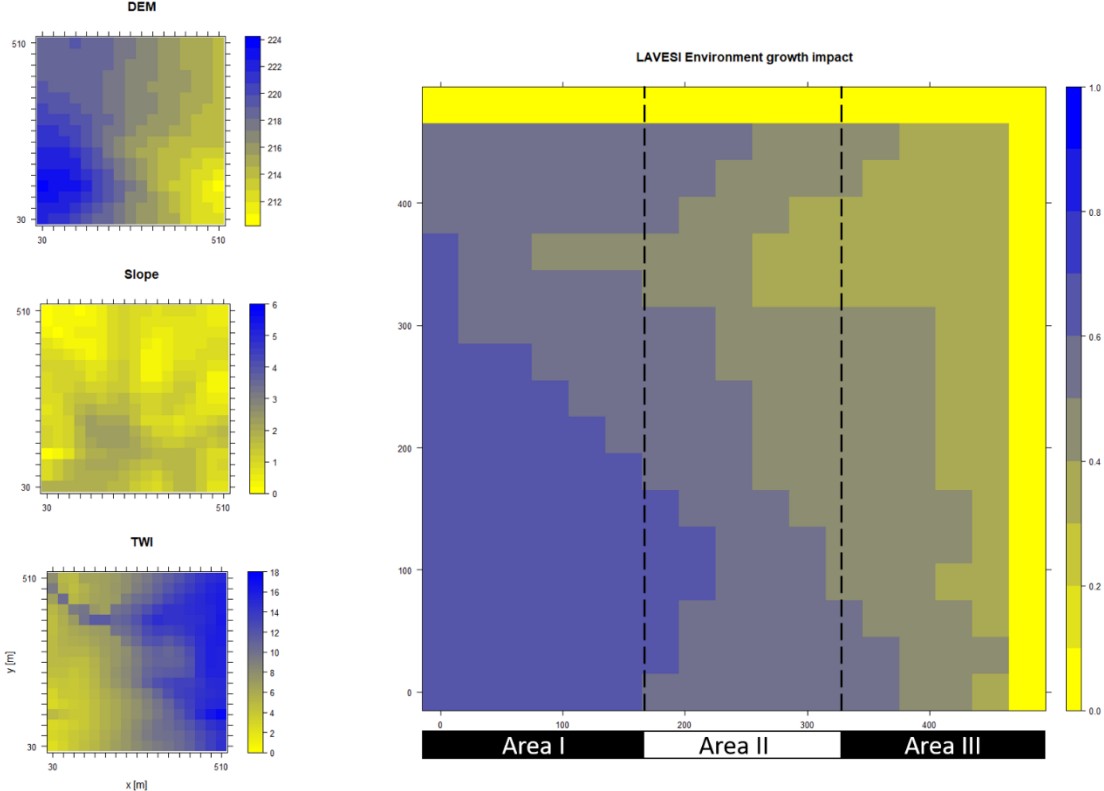

**Figure B3.** Elevation (DEM), slope angle, and terrain water index (TWI) define the environment growth impact (0 no growth possible; 1 good, no constraints) using an empirically fitted function for present forest growth at area of interest Spasskaya Pad.

## Appendix C. CryoGrid model parameters and constants used

**Table C1. Overview of the CryoGrid parameters used.**

| Process / Parameter | | Value | Unit | Source |
|---|---|---|---|---|
| Density falling snow | $\rho_{snow}$ | 100 (SPA), 200 (NYU/KHA) | kg m$^{-3}$ | Stuenzi et al. (2021a) |
| Albedo ground | $\alpha$ | 0.3 | - | Stuenzi et al. (2021a) |
| Roughness length | $z_0$ | 0.001 | M | Westermann et al. (2016) |
| Roughness length snow | $z_{0snow}$ | 0.0001 | M | Boike et al. (2019) |
| Geothermal heat flux | $F_{lb}$ | 0.05 | W m$^{-2}$ | Westermann et al. (2016) |
| Thermal cond. mineral soil | $k_{mineral}$ | 3.0 | W m$^{-1}$ K$^{-1}$ | Westermann et al. (2016) |
| Emissivity | | 0.99 | - | Langer et al. (2011) |
| Root depth | | 0.2 | M | Stuenzi et al. (2021a) |
| Evaporation depth | | 0.1 | M | Nitzbon et al. (2019) |
| Hydraulic conductivity | | 10-5 | m s$^{-1}$ | Boike et al. (2019) |

**Appendix D. Spatial distribution of the leaf area index (LAI) for mixed species and pure species simulations at the focus regions in 5-year steps.**

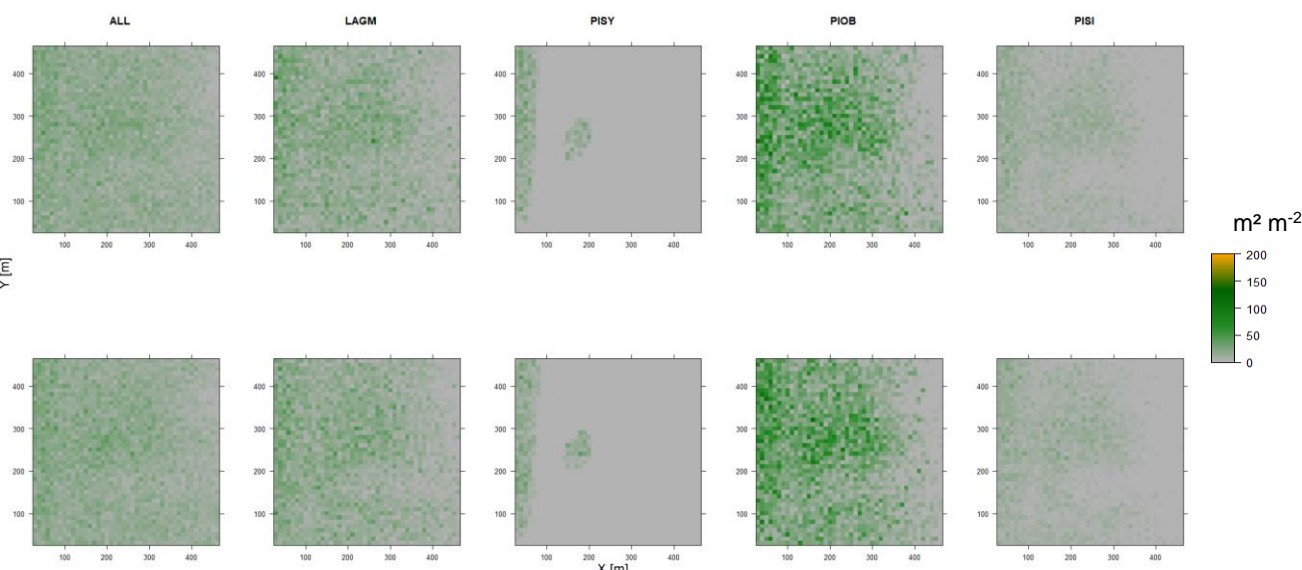

**Figure D1. Leaf area index (LAI) values of the CryoGrid-grid aggregated at year 2015 at Lake Khamra. Upper row LAVESI-CryoGrid coupled; lower row LAVESI simulations. LAGM *Larix gmelinii*; PIOB *Picea obovata*; PISI *Pinus sibirica*; PISY *Pinus sylvestris*.**

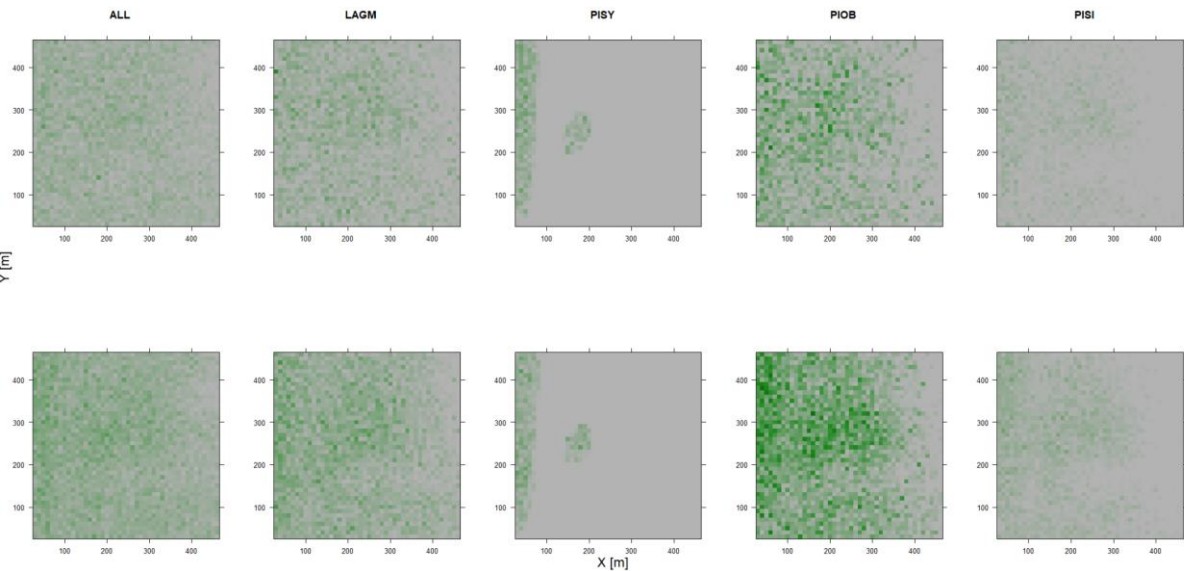

**Figure D2. Leaf area index (LAI) values of the CryoGrid-grid aggregated at year 2020 at Lake Khamra. Upper row LAVESI-CryoGrid coupled; lower row LAVESI simulations. LAGM *Larix gmelinii*; PIOB *Picea obovata*; PISI *Pinus sibirica*; PISY *Pinus sylvestris*.**

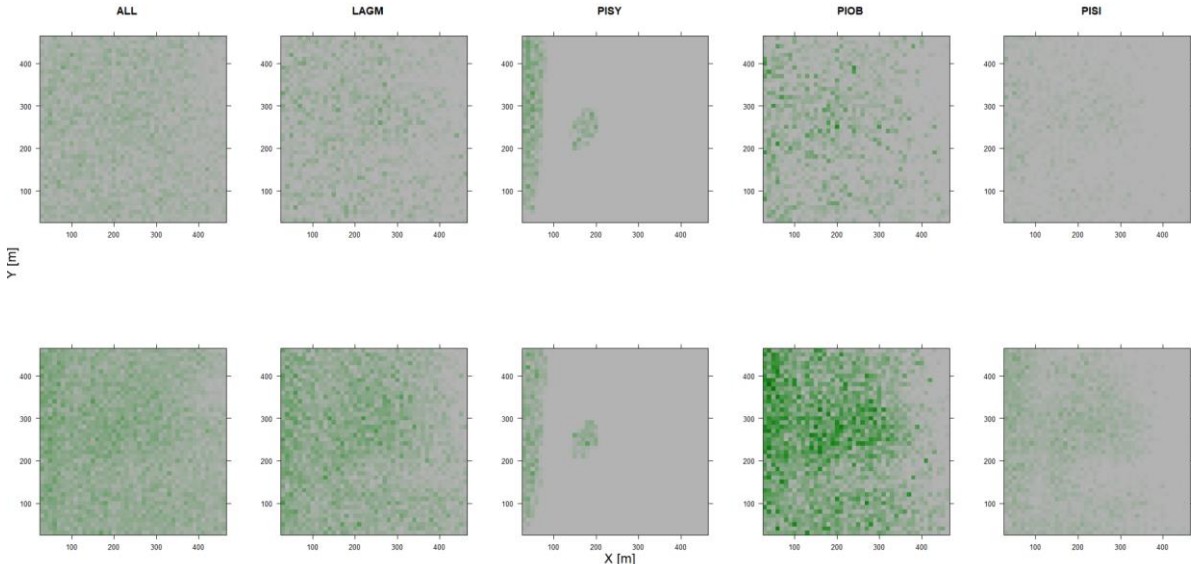

**Figure D3. Leaf area index (LAI) values of the CryoGrid-grid aggregated at year 2025 at Lake Khamra. Upper row LAVESI-CryoGrid coupled; lower row LAVESI simulations. LAGM *Larix gmelinii*; PIOB *Picea obovata*; PISI *Pinus sibirica*; PISY *Pinus sylvestris*.**

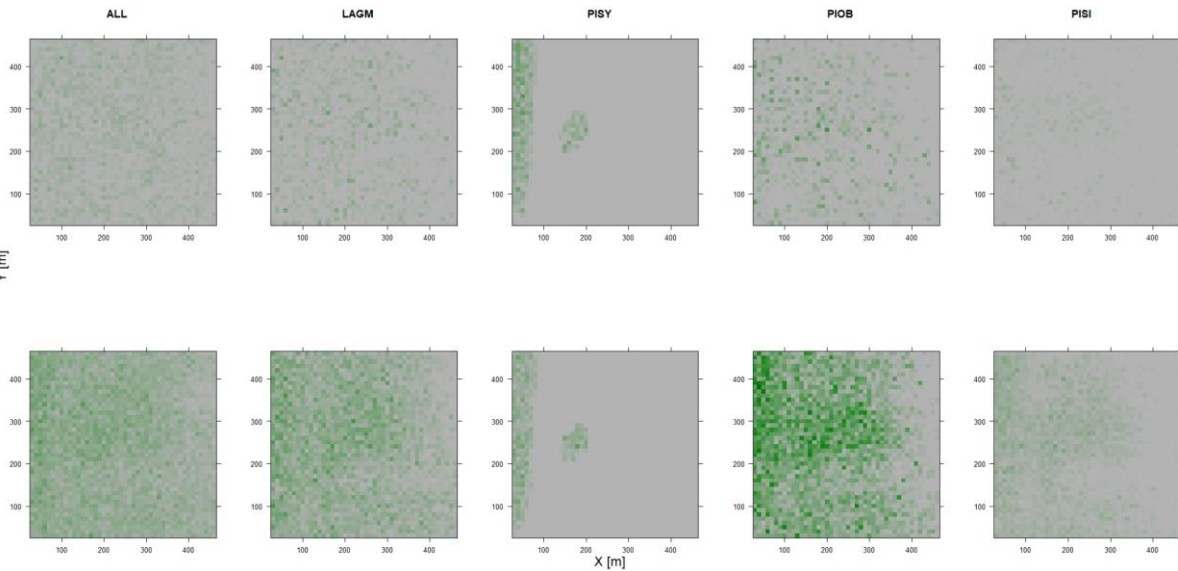

**Figure D4. Leaf area index (LAI) values of the CryoGrid-grid aggregated at year 2030 at Lake Khamra. Upper row LAVESI-CryoGrid coupled; lower row LAVESI simulations. LAGM *Larix gmelinii*; PIOB *Picea obovata*; PISI *Pinus sibirica*; PISY *Pinus sylvestris*.**

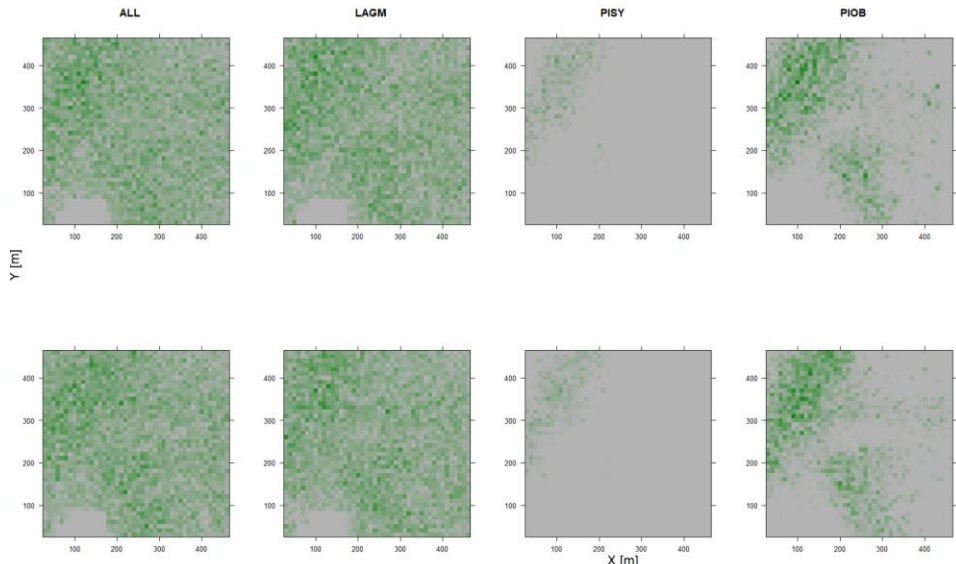

**Figure D5.** Leaf area index (LAI) values of the CryoGrid-grid aggregated at year 2015 at Nyurba. Upper row LAVESI-CryoGrid coupled; lower row LAVESI simulations. LAGM *Larix gmelinii*; PIOB *Picea obovata*; PISI *Pinus sibirica*; PISY *Pinus sylvestris*. Simulations with PISI were not possible.

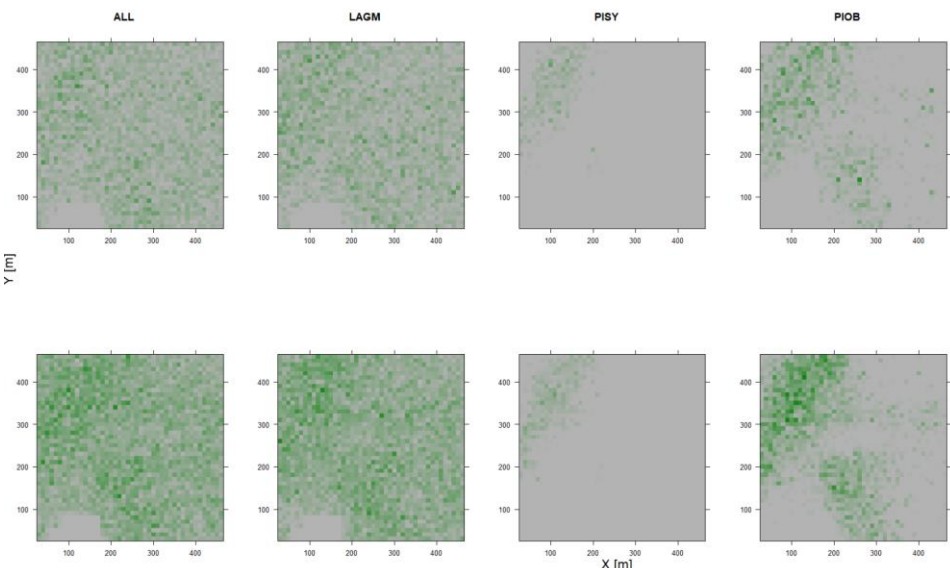

**Figure D6.** Leaf area index (LAI) values of the CryoGrid-grid aggregated at year 2020 at Nyurba. Upper row LAVESI-CryoGrid coupled; lower row LAVESI simulations. LAGM *Larix gmelinii*; PIOB *Picea obovata*; PISI *Pinus sibirica*; PISY *Pinus sylvestris*. Simulations with PISI were not possible.

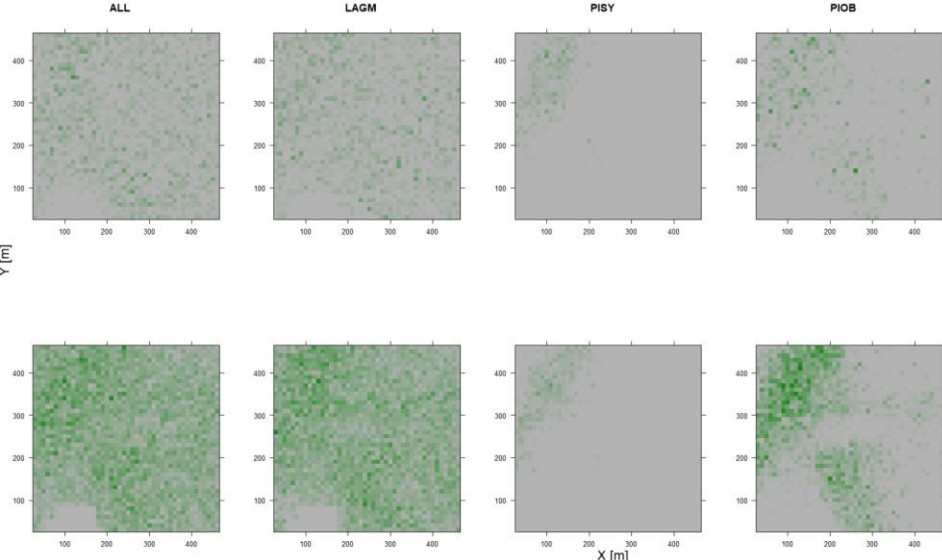

**Figure D7. Leaf area index (LAI) values of the CryoGrid-grid aggregated at year 2025 at Nyurba. Upper row LAVESI-CryoGrid coupled; lower row LAVESI simulations. LAGM *Larix gmelinii*; PIOB *Picea obovata*; PISI *Pinus sibirica*; PISY *Pinus sylvestris*. Simulations with PISI were not possible.**

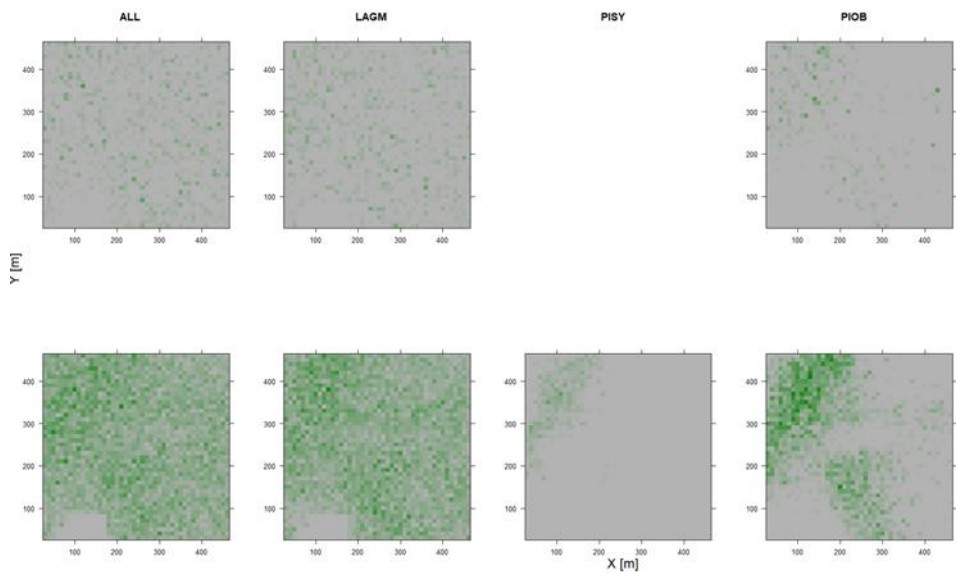

**Figure D8. Leaf area index (LAI) values of the CryoGrid-grid aggregated at year 2030 at Nyurba. Upper row LAVESI-CryoGrid coupled; lower row LAVESI simulations. LAGM *Larix gmelinii*; PIOB *Picea obovata*; PISI *Pinus sibirica*; PISY *Pinus sylvestris*. Simulations with PISI and PISY coupled were not possible.**

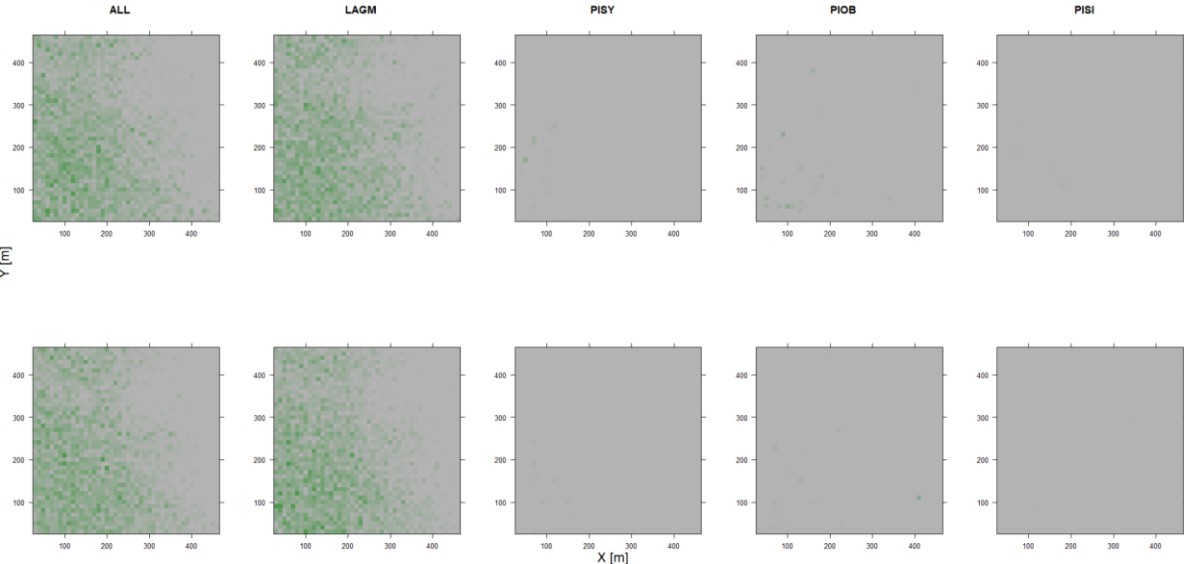

**Figure D9.** Leaf area index (LAI) values of the CryoGrid-grid aggregated at year 2015 at Spasskaya Pad. Upper row LAVESI-CryoGrid coupled; lower row LAVESI simulations. LAGM *Larix gmelinii*; PIOB *Picea obovata*; PISI *Pinus sibirica*; PISY *Pinus sylvestris*.

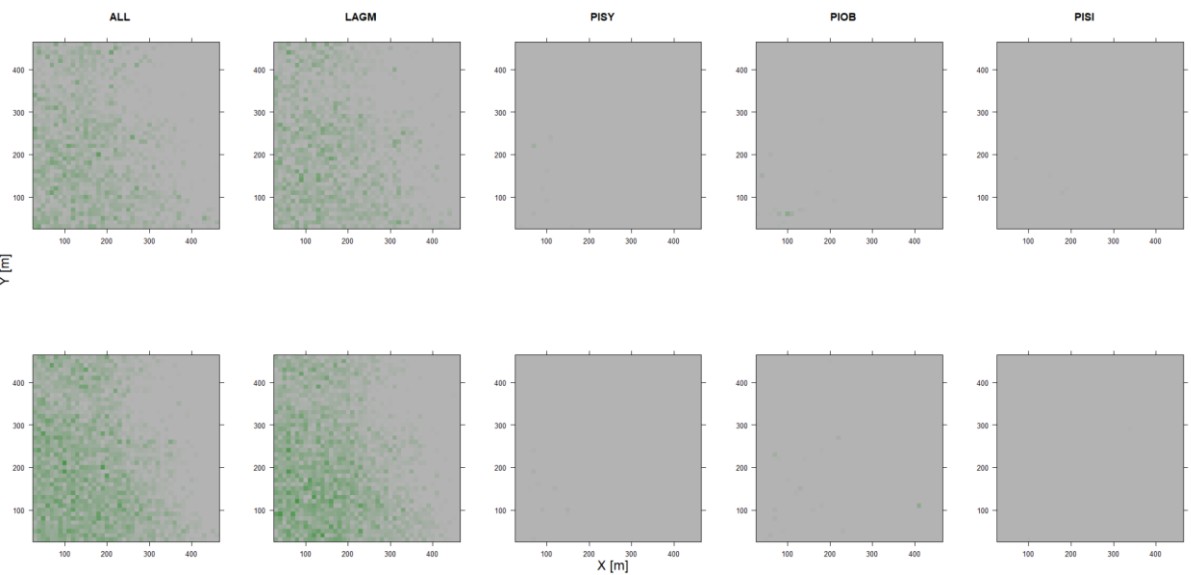

**Figure D10.** Leaf area index (LAI) values of the CryoGrid-grid aggregated at year 2020 at Spasskaya Pad. Upper row LAVESI-CryoGrid coupled; lower row LAVESI simulations. LAGM *Larix gmelinii*; PIOB *Picea obovata*; PISI *Pinus sibirica*; PISY *Pinus sylvestris*.

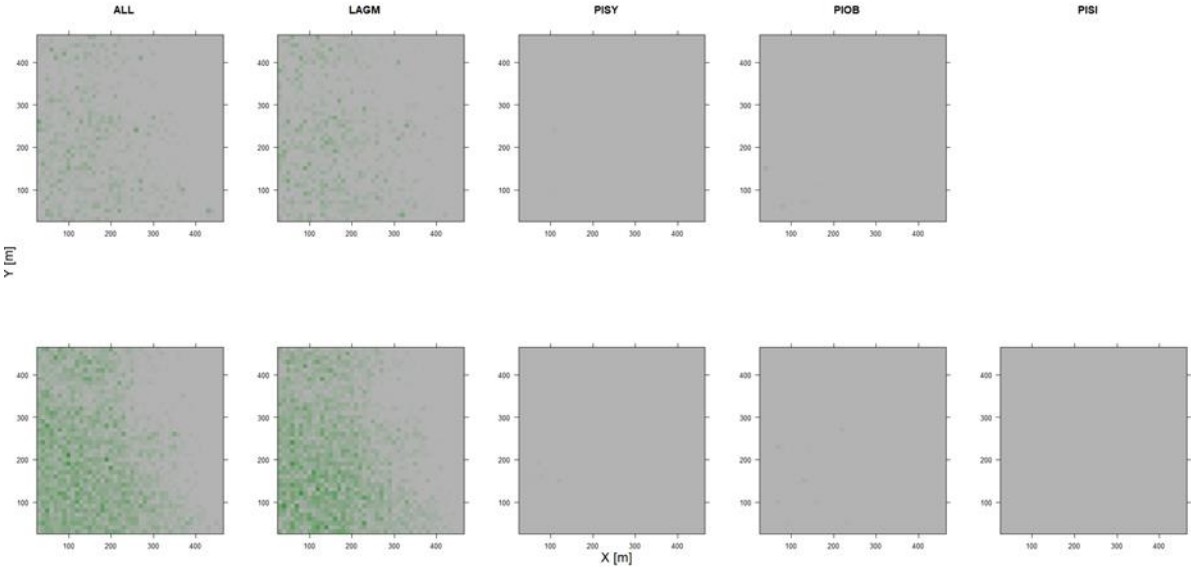

**Figure D11. Leaf area index (LAI) values of the CryoGrid-grid aggregated at year 2025 at Spasskaya Pad. Upper row LAVESI-CryoGrid coupled; lower row LAVESI simulations. LAGM** *Larix gmelinii***; PIOB** *Picea obovata***; PISI** *Pinus sibirica***; PISY** *Pinus sylvestris***. Simulations with PISI coupled was not possible.**

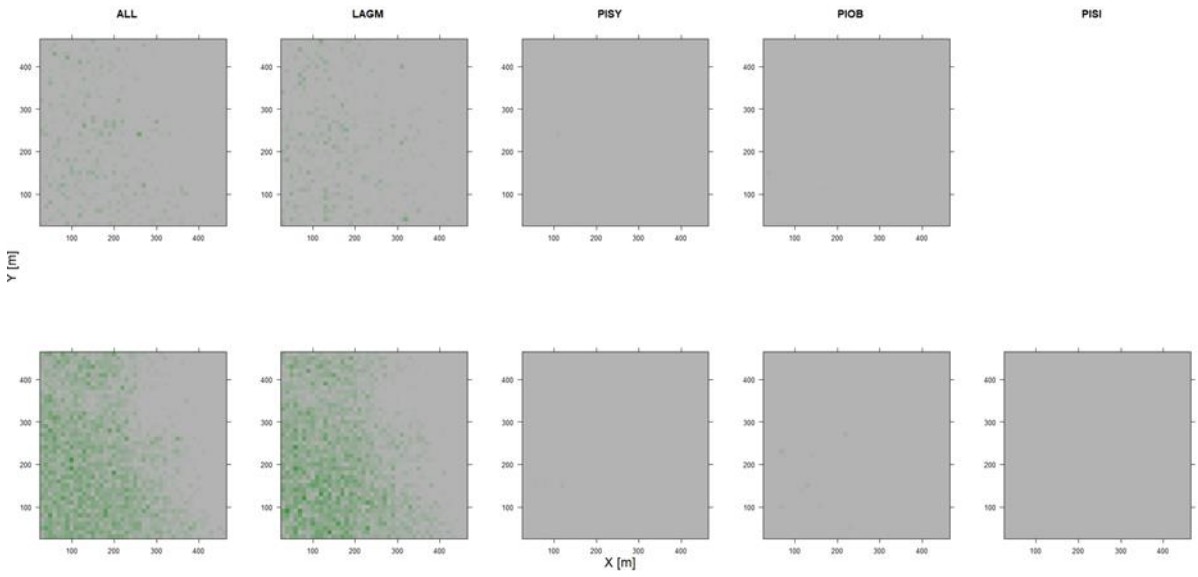

**Figure D12. Leaf area index (LAI) values of the CryoGrid-grid aggregated at year 2030 at Spasskaya Pad. Upper row LAVESI-CryoGrid coupled; lower row LAVESI simulations. LAGM** *Larix gmelinii***; PIOB** *Picea obovata***; PISI** *Pinus sibirica***; PISY** *Pinus sylvestris***. Simulations with PISI coupled was not possible.**

**Appendix E. Evolution of Leaf Area Index (LAI) across the focus region simulation areas**

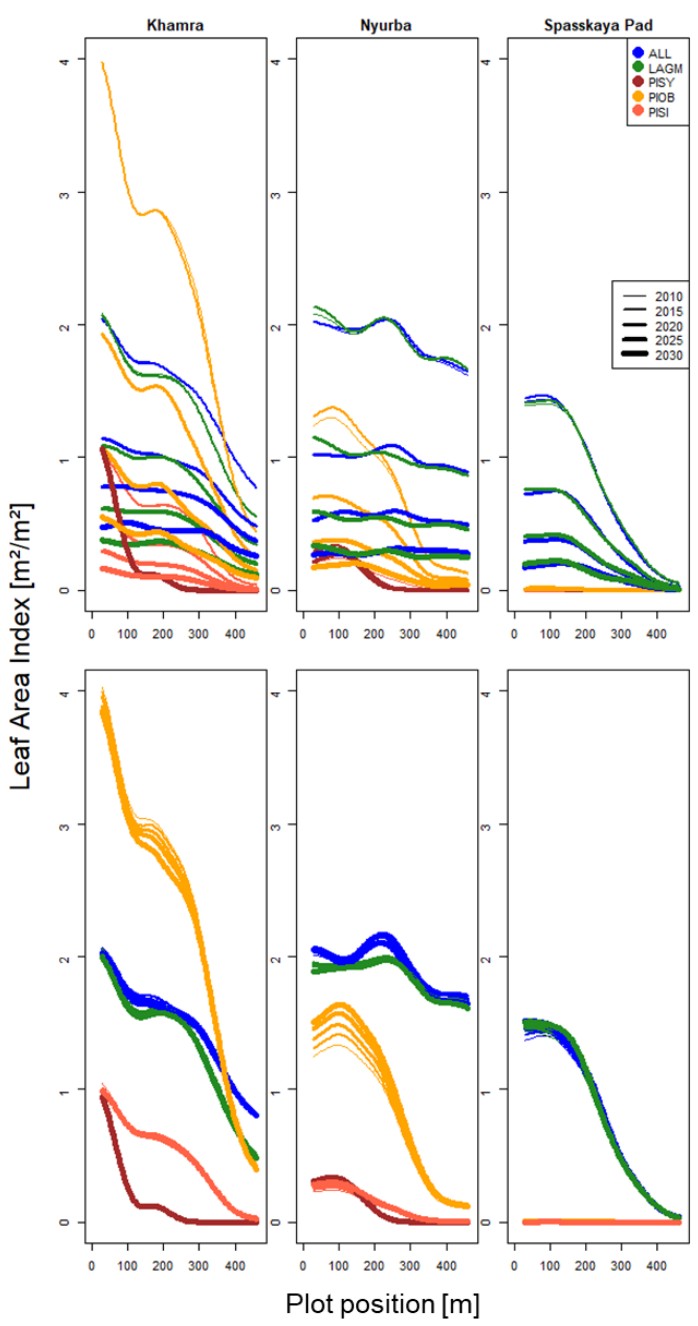

**Figure E1. Mean leaf area index aggregated east-west for each simulated focus area and time slice (2010–2030). Upper row LAVESI-CryoGrid coupled version; lower row only LAVESI.**