# Peer review of "Novel coupled permafrost-forest model (LAVESI-CryoGrid v1.0) revealing the interplay between permafrost, vegetation, and climate across eastern Siberia"

_Geoscientific Model Development, 2021_

## Author Comment (AC1)

Dear Hisashi Sato,

We thank all reviewers for their critical comments and suggestions that led to an improved version of the manuscript. Please find below our point-by-point response to the executive editor comment of Astrid Kerkweg and the two reviewer comments in bold font.

Thank you for handling our manuscript submission and considering the manuscript for final publication.

Kind regards,

Stefan Kruse

**CEC1**: 'Comment on gmd-2021-304', Astrid Kerkweg, 03 Nov 2021

Dear authors,

in my role as Executive editor of GMD, I would like to bring to your attention our Editorial version 1.2: https://www.geosci-model-dev.net/12/2215/2019/

This highlights some requirements of papers published in GMD, which is also available on the GMD website in the 'Manuscript Types' section: http://www.geoscientific-model-development.net/submission/manuscript_types.html

In particular, please note that for your paper, the following requirement has not been met in the Discussions paper:

- "The main paper must give the model name and version number (or other unique identifier) in the title."

Please add the model name (CryoGrid) and a respective version number in the title upon your revised submission to GMD.

Yours,

Astrid Kerkweg

> ***Response:*** ***Thank you for pointing this out. Accordingly, we included in the title now the model name and version number, and tagged the corresponding commit on github as well, which is now also added in the Data Availability section.***
>
> ***The title was changed to:*** *"Novel coupled permafrost-forest model (LAVESI-CryoGrid v1.0) revealing the interplay between permafrost, vegetation, and climate across eastern Siberia"*

**RC1**: 'Comment on gmd-2021-304', Anonymous Referee #1, 11 Nov 2021

The manuscript presents the coupling of an extended version of the model LAVESI, which addressed Siberian larch forests but now can also include further species, with a permafrost model, CryoGrid. The developers of both models are in the authors team. The question addressed is how forest structure and permafrost soil features might change under climate change. The authors show that the coupling with the CryoGrid improves the realism of LAVESI.

The topic addressed is important, the manuscript well prepared and written, and the design of the models, their parameterization, and application look sound. I was a reviewer of the original version of LAVESI and was happy to see its improvements.

Both models are complex and getting them parameterized and then linked was a major effort. Describing this effort takes a lot of space and has been done in all detail.

Nevertheless, it might be worthwhile to start the Methods section with a short characterization of both models, for LaVESI including the original one and the revised one. One way to do so could be filling in a Table for both models with the information required by the Overview part of the ODD protocol for describing models. This would allow readers who are less interest in technical details, at least upon first reading, to get a complete overview of the scales, entities, state variables, and processes included in both models. A detailed description of the scheduling of the combined models process would also be helpful, to better see via which variables both models interact, and how the different temporal resolutions was dealt with. (If all this is described in the Supplement, please refer to the Supplement more explicitly, and more specifically).

> *Response:* ***Following the reviewer's comment, we added a new table in Appendix A as suggested to present the changes from the initial version to the current version presented in this study. In the main text, we included the ODD protocol "Overview" part in the model descriptions and added the suggested short description at the beginning of the methods section.***
> ***This paragraph is added as follows:*** *"We further developed two models and start each description by using the Overview part of the ODD protocol for describing individual based models (Grimm et al, 2010). We describe the host model LAVESI in section 2.1, a spatially-explicit, individual-based model handling the full life-cycle of tree species and interactions among individuals and its environment. The second model CryoGrid, which is informed by the host model and delivers improved state variables back to LAVESI, is described in section 2.2, a one-dimensional, numerical land surface model that simulates the thermo-hydrological regime of permafrost ground by numerically solving the heat-conduction equation."*
>
> ***The second part of the comment about different scales and interaction between models. We described in section 2.3 Coupling the relevant items. For a clarification of the interaction and the scheduling of the two coupled models we improved Fig. 2.***

*The figure was changed to:*

[Figure]

The only thing I stumble over while reading with the 20 cm resolution of the "environmental grid" - where does this resolution come from= From the original DEM?

> *Response:   The 20 cm resolution of the environment grid was chosen and tested in the original setup of the model LAVESI. It allows for interaction of the trees in a resource efficient way rather than computing the interaction among each individual tree pair, which would be not possible on larger areas with several millions of established trees and magnitudes more seedlings. Unfortunately, the DEMs that are freely available have a resolution of 90 or 30 m. To make use of them we interpolate the original data. However, this approach is underestimating the micro-topography of significant changes over tens of centimeters (own observation).*

There are some general comments on the the quality of the model predictions in terms or realism (i.e., validation, corroboration). I think this should made more explicit, and more specifically linked to the results. While there might be no single data set allowing for a comprehensive validation, putting different aspects together might help convincing readers that this not only a technical exercise, but indeed improved the predictive capacity of LaVESI. Perhaps, in the sense of pattern-oriented modelling, another Table might make sense which list which aspects of the model output match observations - even if only in the widest, or qualitative way.

> *Response:   We followed the suggestion of the reviewer to give a clearer overview about the model performance. Therefore, we inserted in the discussion a summary table (Table 4) with expectations based on observations including the references for each item we added the simulation results, both, qualitatively and quantitatively if necessary.*
>
> *Table 4 is as follows:*

*Table 4. Qualitative comparison of simulation results to expectations based on observations.*

| Pattern | Expectation | Simulation study |
|---|---|---|
| ***Species presence*** | • *Khamra:*
• *mixed-species stands, relatively equal contribution of deciduous/evergreen taxa*
• *warm living taxa (PISI) present \**
• *Nyurba:*
• *Mixes-species stands with larch dominance*
• *no warm living taxa (PISI) present \**
• *Spasskaya Pad:*
• *pure larch forests \*\** | • *LAGM and PIOB dominate (LAI ~1.5 m²/m²)*
• *PISI is present in low numbers (LAI ~0.5 m²/m²)*

• *LAGM most dense (LAI ~1.9 m²/m²)*
• *PISI grows in low numbers (LAI ~0.2 m²/m²)*

• *only LAGM grows (LAI ~0.9 m²/m²)* |
| ***Stand densities*** | • *density gradient: Khamra > Nyurba > Spasskaya Pad \*⁺\*\**
• *species mixtures have higher densities \*\*\*\** | • *stand densities slightly smaller at Khamra than Nyurba, lowest at Spasskaya Pad*
• *species mixtures at Nyurba and Spasskaya are slightly denser than in mono-species simulations (2 vs. 1.9 m²/m²)*
• *mono-species PIOB stands yield higher densities at Khamra* |
| ***Stand distribution*** | • *LAGM generalist vs. PIOB and PISY prefer dryer soils. \*\*\*\** | • *increased drought led populations close to extinction*
• *only mono-species PISY stands are rather constant in densities* |

***\* Kruse et al., 2019a; \*\*Ohta et al., 2001, Sugimoto et al., 2002; \*\*\* Mamet et al., 2019; \*\*\*\* Liang et al., 2016***

***Further, we refer in the text of the discussion to this table and adjusted text parts in the reordered discussion section 4.1 Simulation performance, which is changed to:*** *"Species preference matches observations and expectations (Table 4, Kuznetsova et al., 2010). Larches have a wide ecological niche and are widespread (Mamet et al., 2019). They are generalists and best adapted to the harsh Siberian environments that were predominantly wet but are now become drier with global warming (Churakova et al., 2021, Kharuk et al., 2021). Picea obovata grows best in the westernmost, warm areas and reaches larger LAI/biomass than when growing in mixed stands competing with other species (Kharuk et al., 2007). This is as expected and competition, which, as Wieczorek et al. (2017) shows, seems to be a strong factor dampening the response of tree stands when climatic conditions improve. Further, the simulation of denser stands at the Khamra site contradicts the general observation that mixed-species stands are more productive/denser as the niches are occupied (Liang et al., 2016), but depending on the stand structure it could be negative (Zeller et al., 2018) and is in line with the observation of Chen et al. (2003)."*

Overall, I think this is a great contribution to the modelling of Siberian forests under climate change and, hence, to improving predictions relevant for assessing the global carbon cycle.

**RC2**: 'Comment on gmd-2021-304', Anonymous Referee #2, 23 Dec 2021

Dear Editor/Authors

This manuscript describes a coupled model version amalgamating a one-dimensional permafrost-multilayer forest land-surface model (CryoGrid) with an individual-based and spatially explicit forest model (LAVESI). The authors' approach is extremely important for understanding the current environment of permafrost region and predicting the future, because changes in the permafrost environment due to climate change are closely related to changes in vegetation.

This study certainly has merit, though I've got some concerns list below. The main issue in my mind is structure of the methods section. It is unclear for me what new processes have been added or what parameterizations have been conducted in this study. It would be better if the authors organize them clearly and explicitly. For example,  "gdbasalconst", "gdbasalfac" and "gdbasalfacq" , estimated from tree-ring data, are  parameters newly estimated in this study, so (I think) it should be included in Table 1, not Table B1.

> ***Response:  This is in accordance to comments of Reviewer 1 and as suggested we restructured the methods section and added substantial information. Please see the above comment for a full response on the changes we made.***

If the authors could consider above and below specific comments, I believe that it would improve the readability of this manuscript.

Specific Comments:

P1L27 Should it be Eastern Siberia instead of Central Siberia?

P2L2 Should it be Eastern Siberia instead of Central Siberia?

> ***Response:  Thank you for this comment to clarify the region. The model was applied in this study at sites in Central Yakutia belonging to the geographic region of Eastern Siberia. We changed the region statement as suggested.***

P2L4-L5 I think the reader will understand better if the authors provide specific examples of the positive feedback.

> ***Response:  We improved the sentence following the suggestion of the reviewer by including specific examples of the proposed consequences of global warming for boreal forests.***
>
> ***We changed the sentence to:** "Accordingly, wildfires lead to increased greenhouse gas emissions through burning biomass and by deepening of the seasonally thawed layer for decades. The forest transition furthermore reduces the albedo leading to a net positive global warming feedback, which will likely not be offset by increased carbon sequestration of a denser understory vegetation (Bonan, 2008)."*

P4L15 Which parameter in Table 1 could tell us the better low temperature tolerance of the tree species LACA?

> *Response:* **The reference here was wrong and the relevant species trait *janthresholdtemp* was stated in Table B1. We merged now Table B1 into Table 2 which contains now all updated or new model parameter and state variables.**

P4L25 I think an explanation of gnls-function would help the reader understand it better.

> *Response:* **We included a short description of the fitting function in the text.**
>
> > **The sentence was changed to:** *"Tree-ring width data per species were then imported to R using the dplR package (Bunn et al., 2020) and regression models were set up by fitting nonlinear functions using generalized least squares with the gnls-function from the nlme package (Pinheiro et al., 2019)."*

P4L28-L29 "gdbasalconst", "gdbasalfac" and "gdbasalfacq" are now listd in Table B1, not in Table 1. But, I think they should be listed in Table1 because they are newly estimated in this study.

> *Response:* **See above comment, we merged Table B1 into Table 2.**

P8L12-L14 Since the authors did not compare the simulation results with observed data (they are only comparing the simulation results from LAVESI and the coupled model), I don't think the term "overestimation" is appropriate.

> *Response:* **According to the comment we rephrased the sentence to a comparison between the two model variants here in the results section. However, we still think the term "overestimation" is quite suitable in the conclusion section as the model CryoGrid was validated against measurement data; as pointed out in a comment below. Further, we added more information about expectations and a comparison to simulation results.**
>
> > **The sentence was changed to:** *"In nearly all years, LAVESI's ALT values are higher by up to 20 cm (mean over all is 109.6±11.4 cm versus 96.1±10.2 cm, which is ~14.1%) at all focus regions (Fig. 3). The soil moisture anomaly fluctuates around 0% at Lake Khamra, is lower in the coupled model for Nyurba by ~10%, and Spasskaya Pad by ~20% than in simulations using only LAVESI (Fig. 4)."*

P8L15 I don't really understand what the authors are trying to say. I think all we can know from Figure 3-4 is that the simulation results from LAVESI and the coupled model are different.

> *Response:* **We assume that CryoGrid, which was validated against measurement data is achieving more realistic values and hence the coupled model in which LAVESI is using those critical values (ALT and soil humidity) from CryoGrid to be corrected for the error. We deleted the misleading sentence at the end.**

P8L26-L27 What are the factors that cause LAI values to decrease? Could the authors describe possible factors?

>    *Response:*   **We clarified the use of LAI here and in the paragraph before. The LAI is a proxy for stand density, which is calculated from the dimensions of the present individual trees in a certain area.**
>
>    **The sentence was changed to:** *"Smaller population sizes can be observed in all simulations leading to a drop in LAI values when LAVESI is updated by CryoGrid (comparing lower to upper panels in Fig. 5 & 6 & 7)."*
>
>    **And the text in the beginning of first paragraph 3.1 was changed to:** *"A gradient of population densities (expressed in LAI values) forms, which negatively follow the TWI gradient on all sites (I: left, driest to III: right, wettest, Fig. 5 & 6 & 7)."*

P9L11-L12 Reference are required.

>    *Response:*   **We added two recent publications finding increased drought in Siberia in the recent decades.**
>
>    **The sentence was changed to:** *"They are generalists and best adapted to the harsh Siberian environments that were predominantly wet but are now become drier with global warming (Churakova et al., 2021, Kharuk et al., 2021).* Picea obovata *grows best in the westernmost, warm areas and reaches larger LAI/biomass than when growing in mixed stands competing with other species (Kharuk et al., 2007)."*

P22 I don't see Sato et al., 2010 (in table 1) in the list of references; is that a mistake for Sato et al., 2016?

>    *Response:*   **Thank you for the comment, we corrected our mistake and refer now to the correct publication of Sato et al. from the year 2016.**